# A method for low-coverage single-gamete sequence analysis demonstrates adherence to Mendel's first law across a large sample of human sperm

Sara A Carioscia[1†], Kathryn J Weaver[1†], Andrew N Bortvin[1], Hao Pan[1], Daniel Ariad[1], Avery Davis Bell[2], Rajiv C McCoy[1]*

[1]Department of Biology, Johns Hopkins University, Baltimore, United States; [2]School of Biological Sciences, Georgia Institute of Technology, Atlanta, United States

**Abstract** Recently published single-cell sequencing data from individual human sperm (n=41,189; 969–3377 cells from each of 25 donors) offer an opportunity to investigate questions of inheritance with improved statistical power, but require new methods tailored to these extremely low-coverage data (~0.01× per cell). To this end, we developed a method, named rhapsodi, that leverages sparse gamete genotype data to phase the diploid genomes of the donor individuals, impute missing gamete genotypes, and discover meiotic recombination breakpoints, benchmarking its performance across a wide range of study designs. We then applied rhapsodi to the sperm sequencing data to investigate adherence to Mendel's Law of Segregation, which states that the offspring of a diploid, heterozygous parent will inherit either allele with equal probability. While the vast majority of loci adhere to this rule, research in model and non-model organisms has uncovered numerous exceptions whereby 'selfish' alleles are disproportionately transmitted to the next generation. Evidence of such 'transmission distortion' (TD) in humans remains equivocal in part because scans of human pedigrees have been under-powered to detect small effects. After applying rhapsodi to the sperm data and scanning for evidence of TD, our results exhibited close concordance with binomial expectations under balanced transmission. Together, our work demonstrates that rhapsodi can facilitate novel uses of inferred genotype data and meiotic recombination events, while offering a powerful quantitative framework for testing for TD in other cohorts and study systems.

*For correspondence:
rajiv.mccoy@jhu.edu

†These authors contributed equally to this work

## Editor's evaluation

The paper reports a method to study deviations from Mendelian inheritance in genomic data from gametes. The authors use this method to study the existence of the phenomenon in human sperm data but do not find it. The method will be useful for future studies on segregation distortion, and the findings are an important step for the systematic study of segregation distortion in humans and other organisms.

## Introduction

The recent development of high-throughput single-cell genome sequencing of human sperm (termed "Sperm-seq") (**Bell et al., 2020**; **Leung et al., 2021**) offers an opportunity to study various aspects of meiosis and inheritance with improved statistical power. Using a highly multiplexed droplet-based approach, Sperm-seq facilitated sequencing of thousands of sperm from each of 25 donor individuals (n=41,189 total cells), in turn revealing detailed patterns of meiotic recombination and aneuploidy.

**eLife digest** Many species on Earth can carry up to two different versions of a given gene, with each of these 'alleles' having only a 50/50 chance of being transmitted to the next generation via sexual reproduction. Certain 'selfish' sequences, however, can hijack this process and increase their probability of being passed on to an offspring. Known as transmission distortion, this phenomenon may result in alleles spreading through the population even if they are detrimental to fertility.

Transmission distortion has been detected in many species such as flies, mice and some plants. It can take place at various stages during reproduction; for example, the selfish alleles may become overrepresented among eggs or sperm. However, scientists need to study a large number of offspring or reproductive cells to be able to detect whether an allele is inherited more often than expected. This has made it difficult to determine whether transmission distortion also happens in humans, and research so far has resulted in conflicting conclusions.

A recently published dataset of human sperm from 25 donors offered Carioscia, Weaver et al. the opportunity to examine this question. Every volunteer had produced between 969 and 3377 sperm cells, each with about 1% of their genome sequenced. Carioscia, Weaver et al. developed a computational method, which they named rhapsodi, that allowed them to 'fill in the gaps' and infer missing regions of the genome for each cell. To do so, they relied on the fact that sperm cells from a given individual are highly related to one another.

With this more complete data at hand, it became possible to look for evidence of transmission distortion by searching for alleles that were overrepresented in sperm from a given donor. No selfish sequence could be detected in any of the 25 individuals, suggesting that human sperm may not be subject to pervasive transmission distortion. Signatures of selfish alleles detected in previous human studies may have therefore not resulted from this mechanism taking place at the sperm level. Instead, transmission distortion in humans could primarily target eggs or happen at later stages (for instance, if embryos carrying the selfish allele have better chances of survival).

The 'rhapsodi' method developed by Carioscia, Weaver et al. should allow other scientists to work with datasets for which large portions of the genetic information is missing. It may therefore become easier for researchers to track selfish alleles which are difficult to detect, and to examine bigger, more diverse samples which also include individuals with known fertility challenges.

However, the low sequencing coverage per cell (~0.01×) necessitates the development of tailored statistical methods for recovering gamete genotypes.

To this end, we developed a method called rhapsodi (R haploid sperm/oocyte data imputation) that uses low-coverage single-cell DNA sequencing data from large samples of gametes to reconstruct phased donor haplotypes, impute gamete genotypes, and map meiotic recombination events (*Figure 1*). Here, we introduce this method and quantify its performance over a broad range of gamete sample sizes, sequencing depths, rates of recombination, and rates of genotyping error. Key improvements to the haplotype phasing and crossover calling methods from the Sperm-seq paper (*Bell et al., 2020*) include evaluating model performance over a wide range of possible study designs, directly comparing our method to an existing tool, and offering a thoroughly documented and accessible software package.

We then used the resulting imputed genotype data to test adherence to expected rules of inheritance. Specifically, in typical diploid meiosis, each gamete randomly inherits one of two alleles from a heterozygous parent—a widely supported observation that forms the basis of Mendel's Law of Segregation. However, many previous studies have also uncovered notable exceptions, collectively termed 'transmission distortion' (TD), whereby "selfish" alleles cheat this law to increase their frequencies in the next generation. Indeed, examples of TD have been characterized in nearly all of the classic genetic model organisms, as well as numerous other systems (*Fishman and McIntosh, 2019*; *Koide et al., 2012*; *Kim et al., 2014*; *Xu et al., 2014*; *Tang et al., 2013*; *Yu et al., 2011*; *Hulse-Kemp et al., 2015*; *Dai et al., 2016*; *Larracuente and Presgraves, 2012*; *Mcdermott and Noor, 2012*; *Reinhardt et al., 2014*; *Eversley et al., 2010*; *Casellas et al., 2012*; *Wei et al., 2017*). Mechanisms include meiotic drive (*Kursel and Malik, 2018*), gamete competition or killing (*Bravo Núñez et al., 2020*), embryonic lethality (*Bikard et al., 2009*), and mobile element insertion (*Ross and Shoemaker, 2018*).

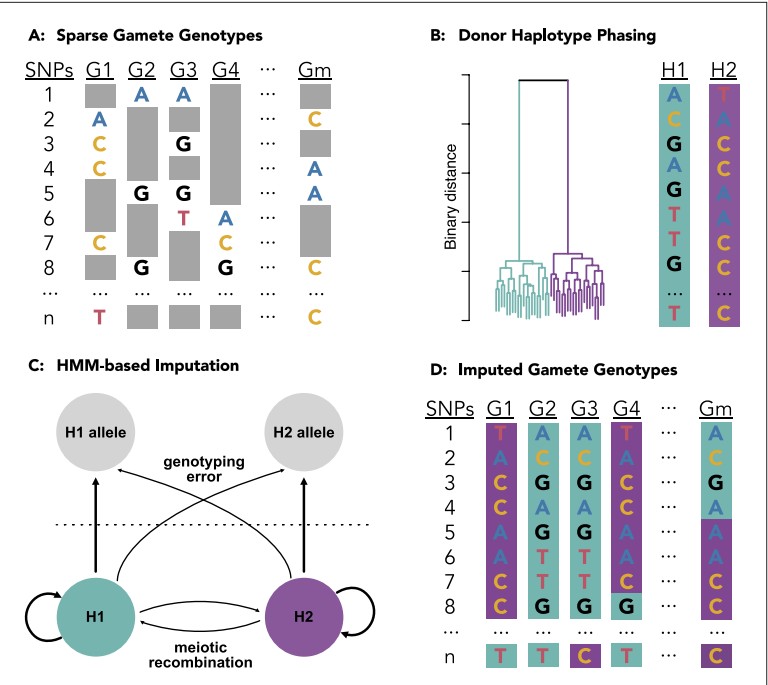

**Figure 1.** Schematic of the methods underlying rhapsodi (R haploid sperm/oocyte data imputation). Low coverage data from individual gametes (**A**) is clustered to phase the diploid donor haplotypes (**B**). A Hidden Markov Model, with tunable rates of genotyping error and meiotic crossover, is applied to trace the most likely path along the phased haplotypes for each gamete (**C**) thereby imputing the missing gamete genotypes (**D**) which can be used to discover meiotic recombination events as transitions from one donor haplotype to the other (e.g., purple [H2] to teal [H1] in gamete G4 between SNPs 7 and 8).

Such phenomena are frequently associated with sterility or subfertility (*Schimenti, 2000*; *Higgins et al., 2018*), but may spread through a population despite negative impacts on these components of fitness (*Phadnis and Orr, 2009*).

Previous attempts to study TD in humans have revealed intriguing global signals but did not identify individual loci that achieved genome-wide significance and could be discerned from sequencing or analysis artifacts (*Mitchell et al., 2003*). For example, using data from large human pedigrees, *Zöllner et al., 2004* reported a slight excess of allele sharing among siblings (50.43%)—a signal that was diffuse across the genome, with no individual locus exhibiting a strong signature. *Meyer et al., 2012* applied the transmission disequilibrium test (TDT) (*Spielman et al., 1993*) to genotype data from three large datasets of human pedigrees. While multiple loci exhibited signatures suggestive of TD, the authors could not confidently exclude genotyping errors, and the signatures did not replicate in data from additional pedigrees. Similarly, *Paterson et al., 2009* applied the TDT to large-scale genotype data from the Framingham Heart Study but attributed the vast majority of observed signals to single-nucleotide polymorphism (SNP) genotyping errors.

Analysis of large samples of gametes, either by pooled (*Corbett-Detig et al., 2015*; *Corbett-Detig et al., 2019*) or single-cell genotyping (*Meyer et al., 2012*), offers an alternative approach for discovering TD, albeit only for mechanisms operating prior to the timepoint at which the gametes are collected (e.g., meiotic drive or gamete killing). Many well-characterized instances of TD across organisms relate to male gametogenesis (*Navarro-Dominguez et al., 2022*; *Verspoor et al., 2020*; *Corbett-Detig et al., 2019*), and genotyping of sperm cells allows isolated investigation of this process, without possible opposing effects of selection at later stages. *Meyer et al., 2012*, for example, performed sperm genotyping in the attempted replication of their pedigree-based test. *Wang et al., 2012* and *Odenthal-Hesse et al., 2014* used sequencing and targeted genotyping, respectively, to scan samples of sperm cells for evidence of TD. While neither study observed the long tracts of TD signals that are expected under a classic model of meiotic drive, they did uncover short tracts suggestive of biased gene conversion. This observation is potentially consistent with the reported rapid

evolutionary turnover in the landscape of meiotic recombination hotspots (*Coop and Myers, 2007*). Such hotspots are associated with high rates of crossovers and non-crossovers, given that the repair of meiotic double-strand breaks produces short tracts of gene conversion.

Importantly, previous studies of TD in humans have been limited in their statistical power for detecting weak TD. Power of pedigree-based studies has been constrained by the small size of human families, although power may be gained for common polymorphisms by aggregating signal across multiple families. Gamete-based studies were historically constrained by costs and technical challenges of single-cell genotyping, limiting analysis to relatively few gametes or small portions of the genome. Specifically, previous single-cell studies used sample sizes of fewer than 500 sperm per donor and performed targeted genotyping of specific loci of interest (e.g., as validation of candidate TD hits) (*Meyer et al., 2012*; *Crouau-Roy and Clayton, 2002*).

To address these limitations, we applied rhapsodi to published single-cell sequencing data from 41,189 human sperm (969–3377 cells from each of 25 donors) (*Bell et al., 2020*; *Leung et al., 2021*). After stringent filtering for segmental duplications and other sources of genotyping error, our results exhibited close concordance with null expectations under Mendelian inheritance, both with regard to individual loci and to aggregate genome-wide signal. Our study thus suggests balanced transmission of alleles to the gamete pool in this sample.

## Results

### A method for single-gamete sequencing analysis

We developed a method to phase donor haplotypes, impute gamete genotypes, and discover meiotic recombination events using low-coverage single-cell DNA sequencing data obtained from multiple gametes from a given donor (see Materials and methods; *Figure 1a*). We describe here the default behavior of rhapsodi, but note in later Results sections and in the Materials and methods additional options and arguments available to the user. Briefly, chromosomes are segmented into overlapping windows, and within each window, the sparse gamete genotype observations at detected heterozygous SNPs (hetSNPs) are clustered to distinguish the two haplotypes (i.e., phase the genotypes) of the diploid donor individual (*Figure 1b*). The sequences of alleles

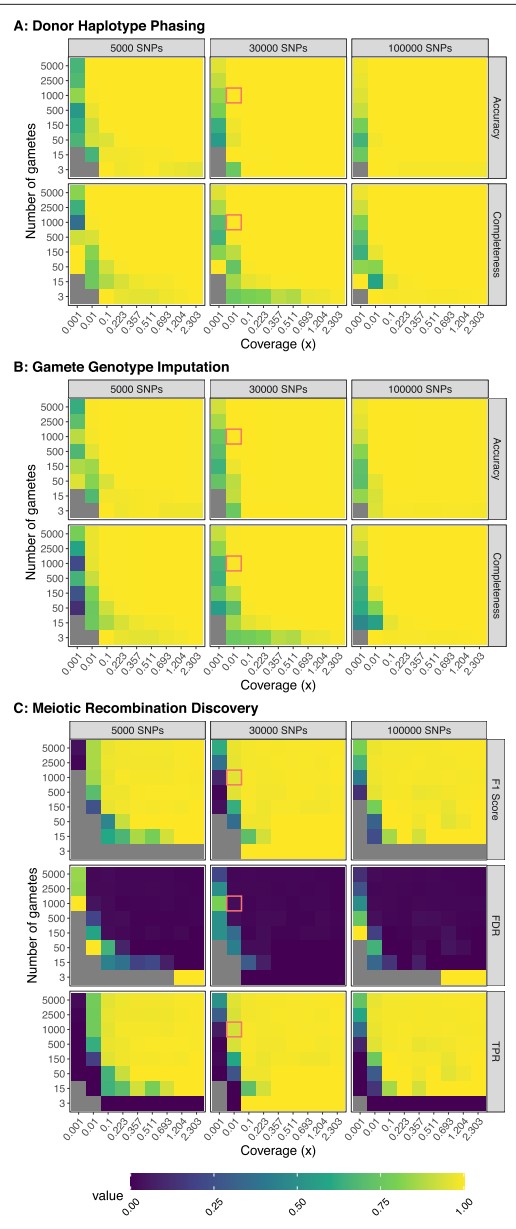

**Figure 2.** Benchmarking performance across a range of study designs. Values represent the average of three independent trials. FDR: False Discovery Rate; TPR: True Positive Rate. For phasing and imputation, gray indicates that no hetSNPs remained after downsampling. For meiotic recombination discovery, gray indicates the absence of a prediction class (e.g., zero FNs, FPs, TNs, or TPs). Simulations roughly matching the characteristics of the Sperm-seq data are outlined in red.

The online version of this article includes the following figure supplement(s) for figure 2:

**Figure supplement 1.** Generative model.

**Figure supplement 2.** Benchmarking performance across a range of study designs - Additional Metrics.

*Figure 2 continued on next page*

*Figure 2 continued*

**Figure supplement 3.** Illustration of switch error rate (SER) and accuracy.

**Figure supplement 4.** Automatic phasing window size calculation.

**Figure supplement 5.** Discovery of meiotic recombination events in simulated data reflecting characteristics of the Sperm-seq data.

**Figure supplement 6.** Model robustness when genotyping error is underestimated.

**Figure supplement 7.** Model robustness when recombination rate is underestimated.

**Figure supplement 8.** Model robustness when genotyping error is overestimated.

**Figure supplement 9.** Model robustness when recombination rate is overestimated.

**Figure supplement 10.** Benchmarking rhapsodi runtime across a range of simulated data profiles.

that compose these haplotypes are decoded based on majority 'votes' within each cluster, and haplotypes from overlapping windows are then stitched together based on sequence identity, thereby achieving chromosome-scale phasing. A Hidden Markov Model (HMM) with (1) emission probabilities defined by rates of genotyping error and (2) transition probabilities defined by rates of meiotic crossover is then used to infer the most likely path along the phased haplotypes for each gamete (*Figure 1c*), thereby imputing missing genotype data (*Figure 1d*). Points where the paths are inferred to transition from one donor haplotype to the other suggest the locations of meiotic recombination events.

## Evaluating performance on simulated data

To benchmark rhapsodi's performance, we developed a generative model to construct input data with varying gamete sample sizes, sequencing depths of coverage, rates of meiotic recombination, and rates of genotyping error (*Figure 2—figure supplement 1*). We then applied rhapsodi to the simulated data, matching input parameters to those used in the simulations (average of 1 recombination event per chromosome and genotyping error rate of 0.005). Phasing was assessed based on accuracy, completeness, switch error rate, and largest haplotype segment (*Figure 2a*, *Figure 2—figure supplement 2a*). Briefly (but see Materials and methods for complete definitions), 'accuracy' is defined as the proportion of positions where the inferred sequence matches the truth sequence. 'Completeness' is defined as the proportion of non-missing genotypes. 'Switch error rate' is defined as the number of tracts of adjacent mismatches between the inferred and truth sequence, divided by the total number of sequence positions (see *Figure 2—figure supplement 3*). 'Largest haplotype segment' is defined as the longest tract of adjacent matches between the inferred and truth sequence.

Across the range of study designs, we observed that phasing performance (of the default method, termed 'windowWardD2') improved with increasing amounts of data (i.e., increased gamete sample size and coverage). For specific scenarios involving low coverages or small numbers of gametes, this relationship was not always monotonic. This suggests that other parameters that we currently hold fixed (e.g., window size used in phasing) may interact and influence performance. We therefore added an option in rhapsodi to optimize window size based on features of the input data (*Figure 2—figure supplement 4*; see Materials and methods section titled 'Automatic phasing window size calculation'). With the exception of very small gamete counts, we observed that phasing performance reaches a plateau at ~0.1× coverage. However, large sample sizes of gametes can compensate for lower coverages, leading to high performance. Qualitatively similar trends were observed for the tasks of imputing gamete genotypes and discovering meiotic recombination breakpoints (*Figure 2b* and *Figure 2c*; *Figure 2—figure supplement 2b* and *Figure 2—figure supplement 2c*).

Discovery of meiotic recombination events exhibited the weakest relative performance among the three tasks (phasing donor haplotypes, imputing gamete genotypes, and discovering meiotic recombination events), although still strong in absolute terms. This is likely due to a combination of (1) this task's dependence on the successful completion of the previous two tasks, (2) the simplifying assumptions employed within the generative model, and (3) the inherent challenge of this task in data-limited scenarios (i.e., those with low coverage or few SNPs). In relation to the last point, we observed that the resolution of inferred meiotic recombination breakpoints (i.e., the length of the genomic intervals to which inferred crossover events could be localized) was strongly associated with the depth of coverage of the input data (*Figure 2—figure supplement 5a*), as well as the density of underlying hetSNPs across the genome. Assuming a pairwise nucleotide diversity of 1 per 1000 basepairs (bp) (*Sachidanandam et al., 2001*) and given that the theoretical limit of resolution is two hetSNPs, we

found that a coverage of 2.3× (i.e., a missing genotype rate of 10%) was required to approach this theoretical limit. Meanwhile, for coverage resembling the Sperm-seq data (*Bell et al., 2020*) (~0.01×), we estimate a median resolution of 167.5 kilobase pair (kbp), in line with empirical observations from the original study (*Bell et al., 2020*).

Formulating discovery of simulated meiotic recombination events as a classification problem where a predicted recombination breakpoint (or lack thereof) could either be a true positive (TP), true negative (TN), false positive (FP), or false negative (FN) prediction (*Figure 2—figure supplement 5b*), we assessed rhapsodi's performance by computing a false discovery rate (FDR), true positive rate (TPR), and F1 Score, as well as several related metrics (*Figure 2c*; *Figure 2—figure supplement 2c*). Briefly (but see Materials and methods for complete definitions), the 'FDR' is the ratio of false predicted recombination breakpoints to the total number of predicted breakpoints. The 'TPR' is the ratio of true predicted breakpoints to total simulated breakpoints. The 'F1 Score' is the harmonic mean of precision (ratio of true predicted breakpoints to total predicted breakpoints) and TPR (also called 'recall'). As was observed for phasing and imputation, meiotic crossover discovery improved as the amount of data increased, as indicated by increasing F1 scores and TPRs and decreasing FDRs.

Through closer investigation of the locations of FP and FN recombination events (*Figure 2—figure supplement 5b*), we identified three typical error modes. Specifically, we attribute the vast majority of FNs to (1) crossovers occurring near the ends of chromosomes or (2) pairs of crossovers occurring in close proximity to one another, especially at low coverages (*Figure 2—figure supplement 5c*). In the case of co-occurring FNs, the genotype data may be too sparse to capture one or more informative markers that flank the recombination breakpoint(s). Notably, such nearby crossovers should be mitigated by the phenomenon of crossover interference (*Broman and Weber, 2000*), which causes crossovers to be spaced farther apart than expected by chance. By simulating crossover locations under a uniform distribution, our benchmarking strategy is thus conservative in regard to this specific error mode (i.e., over-estimating the FN rate). However, our estimates of the FN rate may be under-conservative in regard to the terminal edges of chromosomes, which have been shown to exhibit high rates of recombination in males (*Halldorsson et al., 2019*). A third mode of error, which manifests as pairs of FPs and FNs, owes to slight displacement of the inferred crossover breakpoint (*Figure 2—figure supplement 5c*), which may arise by consequence of premature or delayed switching behaviors of the HMM.

For study designs mirroring the published Sperm-seq data (*Bell et al., 2020*) (1000 gametes, 30,000 hetSNPs, coverage of 0.01×), rhapsodi phased donor haplotypes with 99.993 (±0.003)% accuracy and 99.96 (±0.03)% completeness (*Figure 2a*); imputed gamete genotypes with 99.962 (±0.002)% accuracy and 99.34 (±0.04)% completeness (*Figure 2b*); and discovered meiotic recombination breakpoints with a mean F1 Score of 0.959 (±0.003), a mean FDR of 1.8 (± 0.3)%, and a mean TPR of 93.7 (±0.3)% (*Figure 2c*). Values are reported as the mean, plus or minus one standard deviation.

We next assessed rhapsodi's robustness to parameter mis-specification by altering the recombination and genotyping error rate parameters relative to those used in generating the simulated data. Only one parameter was mis-specified at a time, while the other was matched to the simulation. While our results suggest overall robustness to model mis-specification, we found that underestimating the genotyping error rate or recombination rate (*Figure 2—figure supplement 6* and *Figure 2—figure supplement 7*) had a greater effect on performance than overestimating either of these parameters (*Figure 2—figure supplement 8* and *Figure 2—figure supplement 9*). In practice, such parameters may be informed based on outside knowledge for a given species (e.g., recombination rate ≈ $1 \times 10^{-8}$ per bp for humans) and sequencing platform (e.g., error rate ≈ 0.005 per bp for Illumina short-read sequencing; *Lou et al., 2013*).

rhapsodi is designed to work for large existing datasets such as Sperm-seq (*Bell et al., 2020*; *Leung et al., 2021*) and to remain applicable as future single-gamete sequencing datasets grow in size. Specifically, rhapsodi was rigorously benchmarked for datasets containing up to 5000 gametes per donor with 100,000 SNPs per chromosome at coverages up to 2.3× (*Figure 2*). However, rhapsodi is capable of analyzing much larger datasets, as we demonstrate through its successful application to simulated data comprising 32,900 gametes (0.01× coverage, 90,795 SNPs) in under 24 hr, multi-threaded on 48 CPU cores (*Figure 2—figure supplement 10*). This represents a dataset ~20× the size of that produced by *Bell et al., 2020*.

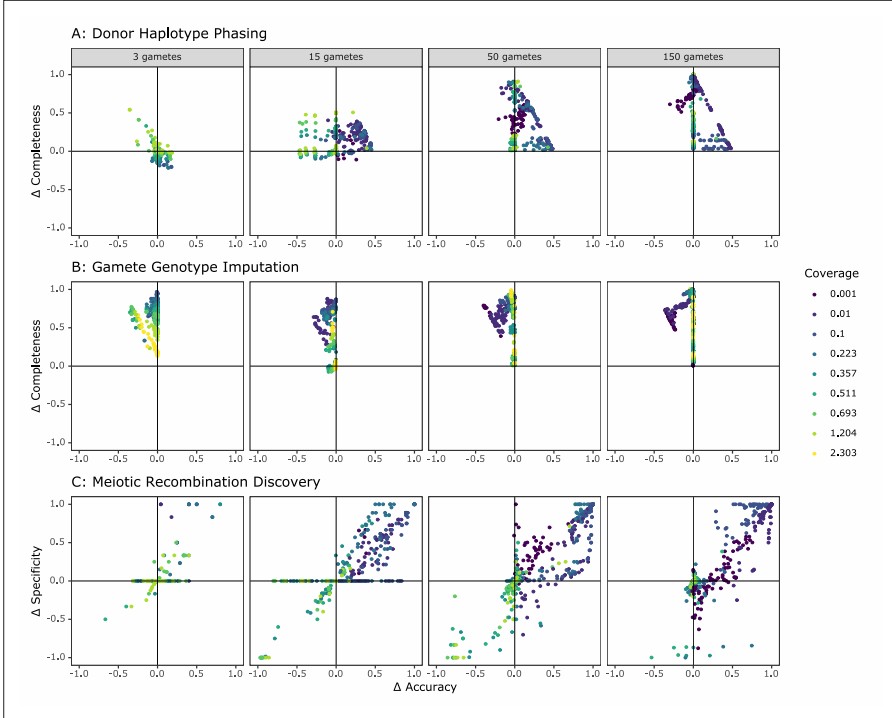

**Figure 3.** Comparison of the performance of rhapsodi to Hapi, an existing gamete genotype imputation tool. For each panel, we depict the difference in performance between the tools (rhapsodi minus Hapi). Each point represents a simulated dataset, and only datasets successfully analyzed by both tools are displayed.

The online version of this article includes the following figure supplement(s) for figure 3:

**Figure supplement 1.** Comparison of the percentage of simulated datasets successfully analyzed by rhapsodi and Hapi across: (1) data import (gmt_import), (2) donor haplotype phasing (phasing), (3) gamete genotype imputation (imputation), and (4) meiotic recombination discovery (CO) stages.

**Figure supplement 2.** Performance of rhapsodi on each simulated dataset, colored based on Hapi's ability to successfully analyze each given data set.

**Figure supplement 3.** Comparison of accuracy and completeness for rhapsodi phasing methods windowWardD2 and linkedSNPHapCUT2.

**Figure supplement 4.** Difference in runtime between windowWardD2 and linkedSNPHapCUT2 algorithms for phasing across a range of simulated data.

## Benchmarking against existing methods: Hapi and HapCUT2

We compared rhapsodi to the existing software tool Hapi, which was previously developed for diploid donor phasing, gamete genotype imputation, and crossover discovery (*Li et al., 2020*), as well as HapCUT2, which was originally developed for read-based phasing of diploid samples (*Edge et al., 2017*) but can be adapted for single-gamete sequence-based phasing using an approach inspired by *Bell et al., 2020*. As the latter approach assumes that alleles originating from a single gamete and chromosome are linked, we hereafter refer to this adaptation as 'linkedSNPHapCut2'.

Hapi was previously shown to outperform the only other haploid-based algorithm, PHMM (pairwise Hidden Markov Model) (*Hou et al., 2013*), as well as two diploid-based phasing methods, What-sHap (*Martin et al., 2016*) and HapCUT2 (standard implementation) (*Edge et al., 2017*) in terms of accuracy, reliability, completeness, and cost-effectiveness (*Li et al., 2020*). While those results were based on different data characteristics than those encountered in our study, we selected Hapi for our phasing, gamete genotype imputation, and crossover discovery comparisons because it was designed specifically for low-coverage gamete imputation, is a reproducible and user-friendly package, and outperformed the existing programs considered in *Li et al., 2020*. We compared the performance of rhapsodi to Hapi using simulated gamete sample sizes ranging from 3 to 150 and depths of coverage ranging from 0.001× to 2.3× (*Figure 3*).

Hapi was designed and optimized for use with low numbers of gametes and was benchmarked using datasets where coverage was greater than 1× (*Li et al., 2020*). Hapi and rhapsodi performed comparably under these conditions (*Figure 3*). Datasets with more than 150 gametes were not possible to benchmark because Hapi's runtimes with larger sample sizes became intractable, taking up to 39 hr per simulated dataset (compared to less than 90 s for rhapsodi; see *Figure 2—figure supplement 10* for a comparison of rhapsodi's runtimes). Of 2916 simulated datasets, Hapi phased, imputed, and detected crossovers in 1902 datasets (65%), while rhapsodi completed the three tasks in 2754 datasets (94%) (*Figure 3—figure supplement 1*). For datasets that Hapi could not analyze, rhapsodi maintained high accuracy and completeness (*Figure 3—figure supplement 2*), with low cost to performance in comparison to datasets that both tools analyzed successfully. Hapi typically achieved high phasing accuracy, but often at the cost of completeness (*Figure 3a*). In contrast, rhapsodi exhibited relative balance between accuracy and completeness. Across a large range of study designs, rhapsodi phased and imputed a greater proportion of SNP genotypes than Hapi with little cost to accuracy (*Figure 3a*, *Figure 3b*). This improvement in completeness was most pronounced at the low coverages (<0.1×) that characterize the Sperm-seq data (*Figure 3*).

While Hapi was previously shown to outperform HapCUT2 (original implementation) in phasing single-gamete genomes (*Li et al., 2020*), *Bell et al., 2020* adapted HapCUT for use with single gamete sequencing data by assuming that alleles originating from the same gamete cell and chromosome were linked. We thus developed a pipeline (to which we refer as 'linkedSNPHapCut2') that converts the files into the necessary format and executes HapCUT2, largely following the previously developed pipeline and detailed in Materials and methods (*Bell et al., 2020*; *Edge et al., 2017*). We then benchmarked rhapsodi's default windowWardD2 phasing approach against linkedSNPHapCUT2. The data for these simulations are explained in the Methods section 'Assessing performance with simulation'. We applied linkedSNPHapCUT2 to 5713 simulated datasets, on which it ran successfully in 77% of cases, but failed in 21% of cases based on the computing resources available (3 days runtime, 185 Gb memory). Another 1% were unable to be processed due to extreme low coverages (88% below 0.001×) resulting in too few SNPs per chromosome (43 simulations with 1 SNP and 32 simulations with 2–5 SNPs) (*Edge et al., 2017*). Of the simulations that failed with linkedSNPHapCut2, 43% were successfully phased with rhapsodi's default windowWardD2 method.

The major cost of the linkedSNPHapCUT2 approach is the time and memory resources necessary to convert the input data files to the format necessary for use in HapCUT2 (*Figure 3—figure supplement 4*). Because windowWardD2 operates in windows along the chromosome, it runs as a multithreaded process. As such, its overall system time is larger than that of linkedSNPHapCUT2 for datasets with high numbers of gametes, but wall-clock time may be much lower. Both options for phasing within rhapsodi offer high levels of completeness and accuracy across most study designs (i.e., sequencing coverage and number of gametes), including those matching the Sperm-seq dataset (*Bell et al., 2020*). Overall, we find that in data-limited scenarios, linkedSNPHapCUT2 phases haplotypes with a higher completeness than the default windowWardD2 method, but with comparable accuracy (*Figure 3—figure supplement 3*). For higher numbers of gametes, windowWardD2 offers comparable completeness and much higher accuracy (*Figure 3—figure supplement 3*), partially due to the HapCUT2 approach ignoring the positional information that was already encoded in the alignment. In doing so, this method does not take full advantage of the co-inheritance patterns of the SNP alleles, which is an advantage offered by the windowWardD2 approach. Based on these results, we include both windowWardD2 and linkedSNPHapCUT2 in rhapsodi as alternative methods for phasing and recommend use of the latter in scenarios with small numbers of gametes and low coverage.

## Applying rhapsodi to data from single-cell human sperm genomes

Given the strong performance of our method on simulated data, we proceeded to analyze published (*Bell et al., 2020*; *Leung et al., 2021*) single-cell DNA sequencing data from 41,189 human sperm (969–3377 cells from each of 25 donors) (*Figure 5—figure supplement 1*). These data possessed an average sequencing depth of ~0.01× coverage per cell, with a range of ~0.002× to ~0.03×. Of the 25 sperm donors, samples from 20 individuals were obtained from a sperm bank and were of presumed (but unknown) normal fertility status (*Bell et al., 2020*), while five donors had known reproductive issues (failed fertilization after intracytoplasmic sperm injection [*n*=2] or poor blastocyst formation [*n*=3]) (*Leung et al., 2021*). Using principal component analysis, we compared the genetic similarity of

each donor individual to globally diverse populations from the 1000 Genomes Project (*Auton et al., 2015*, *Figure 5—figure supplement 2*). A total of 16 sperm donor individuals clustered with reference samples from the European superpopulation, three donors clustered with reference samples from the East Asian superpopulation, and 2 donors clustered with reference samples from the South Asian superpopulation. Three donors clustered on an axis between the African and European superpopulations, consistent with the admixed African American backgrounds reported at sample collection (*Bell et al., 2020*). Meanwhile, one donor (NC26) showed similarities with both the African and East Asian populations, again consistent with the potential admixed ancestry information reported at sample collection (*Bell et al., 2020*).

Before applying rhapsodi, we performed stringent filtering to mitigate sequencing, alignment, and genotyping errors (see Materials and methods section titled 'Genotype filtering to mitigate spurious TD signatures'). Specifically, we removed low-quality cells and those called as aneuploid for the chromosome of interest; excluded regions of low mappability or extreme repeat content; restricted analysis to known SNPs from the 1000 Genomes Project (*Auton et al., 2015*); and excluded regions exhibiting coverage abnormalities (e.g., suggestive of segmental duplications) in each donor individual's genome (*Figure 5—figure supplement 3*). Applying rhapsodi to these filtered data, we phased donor haplotypes at an average of 99.9 (±0.16)% of observed hetSNP positions per donor and chromosome (i.e. leaving ~0.1% as unknown phase); imputed an average 99.3 (±1.23)% of genotypes per gamete and chromosome; and identified an average of 1.17 (median of 1) meiotic recombination events per gamete and chromosome, with a mean of 25.75 and mode of 24 autosomal crossovers per gamete genome, broadly consistent with the reported lengths of autosomal genetic maps for males (*Broman et al., 1998*; *Halldorsson et al., 2019*). Values are reported as the mean, plus or minus one standard deviation. The average resolution of meiotic recombination breakpoints was 664 kbp (±1.25 Mbp), with a median of 357 kbp, in line with empirical observations from the original study (*Bell et al., 2020*). As previously reported (*Bell et al., 2020*), the inferred crossover locations were concentrated in similar genomic regions across all 25 donors, with the highest densities occurring near the ends of chromosomes (*Figure 5—figure supplement 4*). The overall distributions of crossover locations were

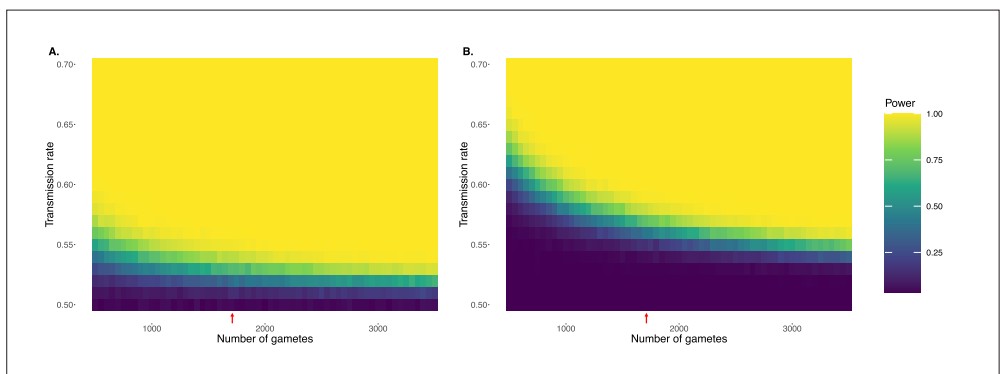

**Figure 4.** Simulation demonstrating power to detect deviations from binomial expectations across sample sizes of sperm, without (**A**) and with (**B**) multiple hypothesis testing correction. For each combination of transmission rate and number of gametes, power was calculated based on 1000 independent simulations and assuming full knowledge of gamete genotypes. Panel A uses the standard $\alpha = 0.05$, while panel B uses an adjusted p-value threshold of $1.78 \times 10^{-7}$ as employed in our study. Note that this correction is conservative in that it adjusts for multiple testing across the genome as well as across donor individuals. Red arrows indicate gamete sample sizes roughly matching the Sperm-seq data (average *n* = 1711 sperm cells per donor).

The online version of this article includes the following figure supplement(s) for figure 4:

**Figure supplement 1.** Simulation demonstrating power to detect deviations from binomial expectations across sample sizes of sperm cells, without (**A**) and with (**B**) multiple hypothesis testing correction.

**Figure supplement 2.** Simulated signature of transmission distortion.

**Figure supplement 3.** Simulated signature of strong (k=0.99) transmission distortion.

**Figure supplement 4.** Simulation demonstrating power to detect deviations from expectations using the transmission disequilibrium test (TDT), as applied to human pedigree studies, without (**A**) and with (**B**) multiple testing correction (alpha = 0.05 and $10^{-7}$, respectively).

qualitatively similar to the patterns observed in the prior analysis of a subset of these donors (*Bell et al., 2020*), as well as strongly correlated with a published male-specific recombination map inferred by trio sequencing of an Icelandic population (*r*=0.9) (*Halldorsson et al., 2019*, *Figure 5—figure supplement 5*). The modest discrepancies between these maps may be driven by a combination of biological (e.g., *PRDM9* genotype, age, etc.) and technical factors—the latter underscored by the observation that the sperm donor sample-specific correlation with the *Halldorsson et al., 2019* map was positively associated with the average depth of coverage of sperm cells from those donors ($\hat{\beta}$ = 5.1, SE = 1.6, p-value = 0.00356).

## Statistical power to detect moderate and strong TD

The scale of the Sperm-seq dataset offers strong statistical power to detect even slight deviations from Mendelian expectations, as supported by our simulations of TD across a range of gamete sample sizes and transmission rates (*Figure 4*, *Figure 4—figure supplement 1*). The 25 donors have an average of 1711 gametes each (range: 969–3377). With this average sample size, we estimate a statistical power of 0.681 to detect deviations of 0.07 (i.e. 57% transmission of one allele in a single donor) and 0.912 to detect deviations of 0.08, accounting for multiple hypothesis testing (p-value threshold = $1.78 \times 10^{-7}$; see below and Materials and methods). For an individual with 950 gametes, we estimate a statistical power of 0.637 to detect deviations of 0.09 and power of 0.84 to detect deviations of 0.1.

Cases of extreme TD pose a potential technical concern, as such loci may appear as homozygous across a sample of sperm, thereby evading detection without outside knowledge of donor hetSNPs. However, our simulations of extreme TD (transmission rate, *k* = 0.99) demonstrate that despite the homozygosity of the causal SNP and nearby SNPs in near-perfect linkage disequilibrium (LD), recombination in flanking regions recovers heterozygosity but still manifests as extreme and detectable TD (*Figure 4—figure supplement 3*). Specifically, across 2200 simulations (100 independent simulations × 22 chromosomes; *k* = 0.99) with parameters matching a typical Sperm-seq donor, we identified the TD signature in all 2200 cases (Power = 1) despite homozygosity (and thus filtering) of the causal SNP in most cases (1958/2200 [89%]). This high power also holds for donor samples with higher (coverage

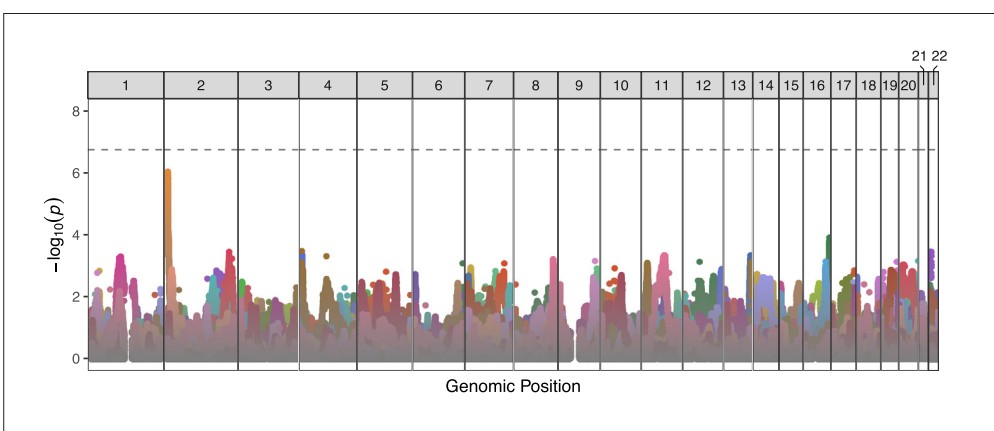

**Figure 5.** Manhattan plot displaying genome-wide TD scan results for all 25 sperm donors across the 22 autosomes. P-values are correlated across large genomic intervals due to the high degree of linkage disequilibrium among sperm cells from a single donor. Colors distinguish results from different donors. No individual test was significant after multiple testing correction (p-value threshold = $1.78 \times 10^{-7}$).

The online version of this article includes the following figure supplement(s) for figure 5:

**Figure supplement 1.** Workflow for application of rhapsodi to data from human sperm and downstream investigation of transmission distortion.

**Figure supplement 2.** Inference of genetic similarity to reference samples from the 1000 Genomes Project.

**Figure supplement 3.** Example of evidence for a segmental duplication in donor NC17, chromosome 6.

**Figure supplement 4.** Recombination map of inferred crossovers in the Sperm-seq data.

**Figure supplement 5.** Comparison of inferred crossover recombination map to published deCODE male-specific recombination map.

**Figure supplement 6.** Visualization of transmission rate of each allele within the pool of sperm, by donor.

= 0.01; Power = 1) and lower (coverage = 0.002; Power = 1) coverages, respectively (see Materials and methods).

## Adherence to Mendelian expectations across sperm genomes

Encouraged by these power simulations, we proceeded to analysis of the Sperm-seq data. Before applying rhapsodi, we performed stringent filtering to mitigate sequencing, alignment, and genotyping errors that could confound our downstream tests of TD, as described above. While our stringent filtering removed a total of 15,138,461 (~30%) SNPs from the input data, we emphasize that true signatures of TD should be minimally affected by such filtering, as such signatures are expected to extend across large genomic intervals due to the extreme nature of LD among the sample of gametes (*Figure 4—figure supplement 2*). For each of the 25 donors, we scanned across the imputed sperm genotypes, performing a binomial test for each detected hetSNP.

Naive multiple testing approaches based on the nominal number of tests would be over-conservative in this context, given the extreme levels of LD across the sample of closely related sperm. We therefore applied a principal component analysis-based approach to infer the effective number of independent statistical tests (*Gao et al., 2008*; *Gao et al., 2010*) and used this value as the basis of a Bonferroni correction that additionally accounted for the multiple testing across donor individuals (p-value threshold = $1.78 \times 10^{-7}$; see Materials and methods). After applying this correction, no individual SNP exhibited evidence of TD at the level of genome-wide significance (*Figure 5*). The distribution of transmission ratio for each allele is shown, by donor, in *Figure 5—figure supplement 6*.

The strongest signal of TD occurred at the end of chromosome 2 (transmission rate = 833 of 1571 [56.2%]; p-value = $9.6 \times 10^{-7}$) for donor NC18 (depicted in dark yellow in *Figure 5*) and encompassed 7 SNPs (rs7603674, rs7578293, rs7567762, rs12478306, rs2084684, rs2385305, rs2100268) within this gene-poor region. These linked SNPs segregate at frequencies of ~0.25 in European populations of the 1000 Genomes Project (*Auton et al., 2015*; *Marcus and Novembre, 2017*; of greatest relevance given the genetic similarities between this donor and individuals from the European superpopulation based on principal component analysis [*Figure 5—figure supplement 2*]).

As noted above, extreme TD (e.g., transmission rate >0.9) could manifest as tracts of homozygosity across the sperm sample at the causal and nearby sites, although TD signal should be detectable in flanking regions, where recombination has restored heterozygosity (*Figure 4—figure supplement 3*). While tracts of apparent homozygosity are indeed observed within the Sperm-seq data (including two donors with tracts exceeding 3.5 Mbp), such tracts are better explained by filtering of technically challenging regions (e.g., low mappability; see *Bell et al., 2020*; Materials and methods), heterozygous deletions, or tracts of homozygosity (e.g., due to identity by descent; *Browning and Browning, 2011*), as in all cases the TD signal at flanking SNPs was unremarkable. Specifically, no site within

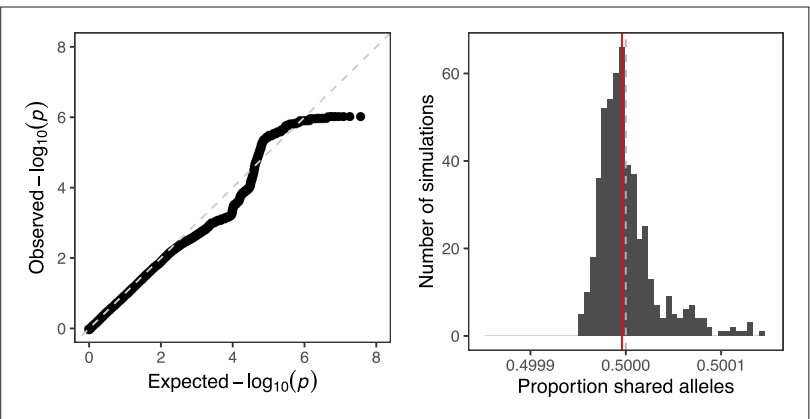

**Figure 6.** Quantifying global evidence of TD. (**A**) Quantile-quantile plot comparing the distribution of p-values from our genome-wide scan for TD to that expected under the null hypothesis. The dashed line corresponds to x = y. (**B**) Mean allele sharing across all pairs of sperm from null simulations (*n*=500) is depicted with grey bars. The same mean computed on the observed data (0.499996) is depicted with a red line, while the null proportion of 0.5 is depicted with a gray dashed line.

1 Mbp of the window of homozygosity yielded a statistically significant result (minimum p-value = 0.072).

## No global signal of biased transmission in human sperm

We also observed that the combined distribution of p-values closely mirrored the expected distribution under the null hypothesis of no TD (*Figure 6*). This absence of global signal is an intriguing contrast to the results of *Zöllner et al., 2004*, especially given the strong statistical power underlying our analysis. We do note that *Zöllner et al., 2004* and other pedigree studies are based on surviving offspring (rather than gametes) and thus may reflect additional sources of TD, as well as selection (see Discussion). To quantify global evidence of TD in our dataset, we calculated the overall degree of allele sharing at observed hetSNPs among sperm from the same donor (*Figure 6*). After pruning the data to SNPs in approximate linkage equilibrium (see Materials and methods section titled 'Calculation of global signal of TD'), we found that across all pairwise comparisons of sperm from each of 25 donors, the average proportion of shared alleles was 0.499996, consistent with the distribution of this proportion under null simulations with no TD (*n*=500 simulations; p-value = 0.47).

## Discussion

Here, we developed a genotype phasing and imputation method, rhapsodi, that leverages sparse gamete genotype data to phase the diploid genomes of the donor individuals, impute missing gamete genotypes, and discover meiotic recombination breakpoints. rhapsodi is available as a thoroughly documented and accessible software package. We benchmarked its performance across a wide range of study designs, with parameters including and much larger than those from existing gamete sequencing datasets.

We then applied rhapsodi to published single-cell sequencing data from individual human sperm (*n*=41,189; 969–3377 cells from each of 25 donors) and tested for TD. In contrast to the tentative locus-specific and genome-wide signals reported in previous studies, our results were consistent with adherence to Mendel's Law of Segregation across this large sample. The signals that we observed are well explained based on the expected variance of the binomial process of Mendelian segregation under the null hypothesis of balanced transmission genome-wide. This negative result hinged on stringent filtering of the input data, as phenomena such as hidden structural variation and ensuing mapping and genotyping errors may otherwise generate false signatures of TD. While TD could of course occur in these filtered regions, it is important to emphasize that even stringent filtering is not expected to eliminate true signals, which should span large genomic intervals due to the extreme LD within the samples of sperm genomes from a given donor (as supported by our simulations as well as empirical data).

One caveat of this and most previous scans for TD in humans is the inability to analyze highly repetitive and other technically challenging regions of the genome, including heterochromatic sequences such as telomeres and centromeres, which have been implicated in TD in other species (e.g., *Brand et al., 2015*; *Axelsson et al., 2010*). Indeed, the strongest signal of TD in our study, albeit below the threshold of genome-wide significance, occurred near the end of chromosome 2. Stringent filtering of repetitive regions was, however, necessary to avoid spurious signatures of TD that may arise due to read mapping and genotyping artifacts (see Materials and methods 'Genotype filtering to mitigate spurious TD signatures'). The recent assembly of a complete human reference genome resolved many heterochromatic sequences for the first time and offers great promise for the future rigorous study of TD genome-wide, especially when combined with other technological advancements such as long-read sequencing (*Nurk et al., 2022*; *Aganezov et al., 2022*).

Past studies of TD in humans have been limited in statistical power, due largely to (1) the challenge of amassing a sufficiently large set of human families (number of families heterozygous at a given site) and number of individuals within a family and (2) the technical limitations of gamete genotyping. Such limitations are compounded by the high burden of multiple testing in genome-wide scans. Importantly, in a single-gamete sequencing study, the number of informative transmissions is equal to the number of genotyped gametes for all observed hetSNPs. In a pedigree-based study, the number of informative transmissions varies across SNPs, as not all parent-offspring trios will include one or more parent heterozygous for a given SNP (*Figure 4—figure supplement 4*). For example,

the Hardy-Weinberg expected proportion of heterozygous parents for a common, autosomal bi-allelic SNP with an allele frequency ($p$) of 0.5 is $2p(1 - p) = 0.5$. Meanwhile, variants with lower frequencies will possess smaller proportions of heterozygotes, thus capturing fewer informative transmissions and limiting statistical power. One implication of this distinction is that pedigree-based studies rely on distorter alleles that act across multiple families and are therefore best suited to discover TD involving variants that are common in the population. Single-gamete sequencing studies require only one individual to be heterozygous for the causal TD allele, although the probability that a donor is heterozygous still depends on the frequency of the allele in the population.

Intuitively, patterns of allelic co-inheritance among single sperm cells from the same donor facilitates the imputation of missing genotype data even at very low depths of coverage (~0.01×) per cell, thereby augmenting our sample size at each heterozygous SNP by approximately 100-fold and improving statistical power compared to naive scans of aligned reads. An alternative but related approach, implemented in several previous studies of non-human species, entails pooled sequencing of gametes (*Corbett-Detig et al., 2015*; *Corbett-Detig et al., 2019*) or offspring individuals (*Wei et al., 2017*). Under the assumption that different gametes are sampled by the reads aligned at each SNP position, TD signal can then be aggregated within localized genomic windows. While this aggregation approach improves statistical power, it sacrifices resolution for identifying the causal locus as well as the ability to simultaneously discover recombination breakpoints. It is also important to note that such aggregation requires external knowledge of the phased haplotypes (e.g., obtained through sequencing of inbred parental strains). Applying a similar approach to pooled sperm sequencing data from humans (e.g., *Breuss et al., 2020*; *Yang et al., 2021*) would thus require additional phasing experiments, either based on the sequencing of relatives, single-molecule long-read sequencing, or patterns of LD in an external reference panel. Formal benchmarking of such strategies and comparison to single-cell methods offers a promising area for future study, and the decision to pursue single versus pooled sequencing likely depends on factors such as library preparation costs and effort, sample availability, and broader goals of the project.

Given the strong statistical power of our study, the absence of TD signal suggests the absence of moderate or strong TD in this sample. This negative result provides an intriguing contrast to the numerous validated examples uncovered in other species. However, our observations do not preclude the occurrence of TD in humans, for reasons that we enumerate here. First, our study was limited to sperm from 25 donor individuals. We would therefore miss TD involving rare or population-specific alleles that may be absent from our sample. A second and related consideration is that if not balanced by countervailing forces, alleles that are subject to TD may rapidly fix in the population, such that detecting these alleles during their ephemeral existence as polymorphisms may be unlikely. Work in model organisms has also demonstrated the possibility that after distorter alleles arise, suppressors may subsequently evolve to re-balance the transmission ratio (e.g., *Greenberg Temin, 2020*). Admixture between populations may then separate the distorter from the suppressor, revealing hidden distorter alleles. A third limitation is that our analysis is solely sensitive to mechanisms of TD that would operate prior to the sampling of the sperm cells (e.g., sperm killing), and is therefore blind to mechanisms that may operate in female meiosis (e.g., meiotic drive) or at different stages of development (e.g., fertilization, maternal-fetal incompatibility, or postzygotic selection). Notably, female meiosis may be particularly susceptible to meiotic drive given the asymmetric nature of the cell divisions that produce the oocyte and polar bodies (*Clark and Akera, 2021*). Within-ejaculate sperm competition (*Sutter and Immler, 2020*) may also contribute to signals of TD observed in other study designs (such as pedigrees) but would not be detected in this study where gametes were sampled indiscriminately. Finally, our analysis does not exclude the existence of TD driven by biased gene conversion, which would generate short tracts below the resolution of our study. Indeed, the occurrence of this phenomenon is well documented based on both comparative evolutionary data and targeted analysis of human recombination hotspots, and is beyond the scope of our study (*Coop and Myers, 2007*).

## Conclusions and future directions

In summary, we introduced and benchmarked a method to phase donor haplotypes, impute gamete genotypes, and infer meiotic recombination events using low-coverage single-cell sequencing data from a large sample of human gametes. Our method is uniquely tailored to extremely sparse sequencing data from large numbers of gametes—a form of data that will be increasingly common as

Sperm-seq and related methods are more broadly adopted and improved. Our simulations demonstrate that rhapsodi outperforms existing methods for phasing and imputation of single-gamete data, while also efficiently scaling to datasets 20-fold the size of that analyzed in our study.

As sequencing large numbers of gametes becomes tenable, even at lower coverages, rhapsodi can facilitate novel uses of inferred genotype data and meiotic recombination events. For example, the ability to construct sex-specific genetic maps could provide essential genetic resources for non-model species to complement genome assemblies and annotations (*Lyu et al., 2021*). In humans, the construction of personalized recombination maps offers opportunities to investigate sources of variability in the recombination landscape, as well as its relation to phenotypes such as fertility status (*Lyu et al., 2021*). For example, whole and partial chromosome gains and losses (i.e., aneuploidies) are the leading cause of human pregnancy loss, and one potential mechanism of aneuploidy formation is the abnormal number or placement of meiotic crossovers (*Lamb et al., 2005*). The chromosome-length haplotypes generated by rhapsodi also offer additional opportunities for population genetic analyses, including inferring kinship and population structure, detecting signatures of positive selection, and assessing demographic history (*Li et al., 2020*).

We applied rhapsodi to scan a large sample of human sperm genomes for evidence of 'selfish' alleles that violate Mendel's Law of Segregation. We observed no significant evidence of TD, either with respect to individual loci or aggregate genome-wide signal, in turn supporting a model of balanced transmission in this sample. Measuring the fidelity of Mendelian segregation and uncovering the mechanisms that safeguard or subvert this process remain important goals for human genetics. Future investigation of TD in humans can employ rhapsodi to examine large samples of sparsely sequenced gametes from diverse individuals and populations—both fertile and infertile. Given the documented fertility impacts of TD in other organisms, a thorough understanding of the loci driving such inheritance patterns could help reveal novel genetic contributions to human infertility. More broadly, our method offers a flexible and reproducible toolkit for testing this and related hypotheses in other cohorts and study systems, especially in light of the declining costs and expanding application of single-cell sequencing methods.

## Materials and methods
### Data input and filtering
The main algorithmic steps of rhapsodi consist of (1) data preprocessing and filtering, (2) reconstruction of phased donor haplotypes (*Figure 1b*), (3) imputation of gamete genotypes (*Figure 1d*), and (4) discovery of meiotic recombination breakpoints. A table of sparse gamete genotypes (SNPs × gametes) with corresponding genome positions is required by rhapsodi as input (*Figure 1a*). Genotypes may be encoded in one of two forms: 0/1/NA (denoting reference, alternative, or missing genotypes, respectively) or A/C/G/T/NA (denoting nucleotides or missing genotypes, respectively). For A/C/G/T/NA encoding, reference and alternative alleles for each position are also required as input. Data is then filtered to consider only SNP positions that are heterozygous for the given individual, with at least one reference and one alternative allele observed across gametes. rhapsodi returns as output phased donor haplotypes, imputed genotype (and corresponding source haplotype) for each SNP in each gamete, and a list of locations of recombination breakpoints detected for each gamete (*Figure 1c*). We detail the methods that produce each output below.

### Reconstruction of phased donor haplotypes
rhapsodi's default windowWardD2 phasing approach leverages patterns of co-inheritance across gametes to reconstruct donor haplotypes from the sparse genotype matrix derived from low coverage sequencing (*Figure 1a*). Chromosomes are divided into overlapping windows; within each window, we compute pairwise binary (Jaccard) distances, cluster the gamete genotypes using Ward's method (*Ward, 1963*) (implemented with the 'ward.D2' option in the `hclust()` function in the R 'stats' package), and cut the resulting tree into two groups to distinguish the haplotypes (*Figure 1b*). For each cluster within a window, we use majority voting to decode the phased haplotype sequences. This approach depends on the reasonable assumption that recombination is sufficiently rare, affecting a minority of gametes within each window. If no genotype achieves a majority at a given position, we designate the phase at that position as ambiguous (denoted as 'NA'). Finally, haplotypes of adjacent

(overlapping) windows are stitched together by considering the genotype consensus (i.e., rate of matching) for the overlapping SNPs. In the stringent stitching mode, if consensus is greater than a given threshold (default = 0.9), the haplotypes under consideration are inferred to be the same (i.e., to derive from the same donor homolog). If the consensus is less than 1 minus the threshold (default = 1 - 0.9 = 0.1), the haplotypes under consideration are inferred to be distinct (i.e., to derive from different donor homologs). Meanwhile, if the consensus lies between these upper and lower boundaries, the windows are considered too discordant to ascertain the relation of the haplotypes, and the method returns an error. However, rhapsodi can optionally be run in a lenient stitching mode which continues with phasing regardless of the level of consensus. In this case, the haplotypes with the highest consensus are assumed to derive from the same homolog.

To summarize, the SNP positions are split into windows. Within each of these windows, binary clustering of the gamete data across SNPs and majority voting are used to reconstruct haplotypes within each window. Finally, adjacent windows are stitched together based on the amount of consensus in the genotypes within the overlapping regions to reconstruct the two haplotypes of the donor genome.

rhapsodi also offers the option to apply a modified version of the diploid read-based phasing approach from HapCUT2 (*Edge et al., 2017*). This includes the functionality necessary for data conversion and calling HapCUT2 within rhapsodi as an alternative phasing approach, termed linkedSNP-HapCUT2. While previous work modified HapCUT for use in haploid phasing (*Bell et al., 2020*), we chose to use HapCUT2, which extends and improves upon the original HapCUT algorithm (e.g., with regard to speed) (*Edge et al., 2017*). To use HapCUT2, the gamete data input to rhapsodi (cell genotype format) is converted into the fragment file format that would be produced as output from the extractHAIRS step if HapCUT2 were applied directly to sample BAM files. This conversion effectively produces synthetic 'fragments' based on the assumption that alleles originating from the same gamete and chromosome are linked.

In our application of rhapsodi to the Sperm-Seq datset, we used the default phasing method (windowWardD2) rather than linkedSNPHapCUT2, due to the poor efficiency of the latter for datasets of this scale. The 0.1% of hetSNP positions that remained of unknown phase were due to ties during the majority vote step.

## Imputation of gamete genotypes

Our imputation approach uses a Hidden Markov Model (HMM) and a subsequent filling step to infer the genotypes of each gamete, effectively tracing the most likely path through the donor haplotypes, given the sparse observations. Our model assumes that (1) the hidden haplotype state at a given SNP depends only on the haplotype state at the immediately preceding SNP, (2) adjacent genotypes originate from the same donor haplotype unless a meiotic recombination event occurred between them, and (3) the observed SNP alleles are either correct or in rare cases arise from genotyping errors (e.g., PCR errors, contaminant DNA co-encapsulated with the sperm genome during library preparation, sequencing errors, mapping errors, etc.). The emission for each position is a single allele, either the H1 allele or the H2 allele (*Figure 1c*). The HMM stays in the current state (either H1 or H2) with some probability and changes state (via meiotic recombination) with some probability. The HMM uses the phased donor haplotypes and accounts for probabilities of genotyping errors and meiotic crossovers: the probability of genotyping errors is reflected in the emission probabilities, while the probability of meiotic crossovers (i.e., the number of expected meiotic recombination events per chromosome per gamete divided by the number of hetSNPs) is reflected in the transition probabilities (*Figure 1c*). We then apply the Viterbi algorithm (*Forney, 1973*) to determine the most likely sequence of hidden states, iterating over all gametes independently (*Figure 1d*). To avoid potential over-smoothing behavior of the HMM, we optionally overwrite the inferred alleles with any observed alleles for rare cases where these genotypes conflict. For example, if the HMM calls a site as deriving from haplotype 1, but the raw data exhibits an allele matching haplotype 2, the method replaces the inferred haplotype 1 allele with the observed haplotype 2 allele.

The HMM considers only observed genotypes and makes no predictions for missing genotypes. To impute missing genotypes, we assume that tracts of unobserved sites that are bordered by SNPs originating from a single donor haplotype originate from that same haplotype, thus filling in the genotypes for these unobserved sites. Similarly, we assume that recombination does not occur between the last observed SNPs and the ends of the chromosomes, again filling in genotypes at unobserved

sites in these terminal regions. However, because of known high rates of recombination at the ends of chromosomes, we provide an option to leave these sites at the ends of chromosomes as unassigned (denoted as 'NA'). Similarly, tracts within recombination breakpoints (i.e., between observed SNPs that transition from haplotype 1 to haplotype 2 or vice versa) remain unassigned (denoted as 'NA').

To summarize, rhapsodi first formulates a Hidden Markov Model (HMM) with emission probabilities defined by the expected genotyping error rate and transition probabilities defined by the expected average recombination rate. The HMM is then fit to the sparse gamete data, tracing the most likely path along the phased donor haplotypes for each gamete. Internal missing genotypes are filled if bordered by matching haplotypes. Missing genotypes at the ends of chromosomes are assigned to haplotypes based on the haplotype state at the first non-missing position or optionally may remain unassigned. Finally, by default, if original genotype observations disagree with inferred genotypes (based on the haplotype state), the original observations are superimposed. These steps together recover the dense gamete genotype matrix.

## Discovery of meiotic recombination breakpoints

Using the imputed gamete genotypes and the underlying phased donor haplotypes, we identified meiotic recombination breakpoints as sites where, for a given gamete chromosome, the inferred path transitions from one donor haplotype to the other. This step should typically be run on the smoothed data (i.e. without superimposing the original genotype observations), as even a low rate of genotyping error could otherwise manifest as false meiotic recombination. Recombination breakpoints are reported as intervals, defined by the last observed site inferred to derive from one haplotype and the first observed site inferred to derive from the other. These start and end points thus demarcate regions in which the meiotic recombination events likely occurred.

## Assessing performance with simulation

We developed a generative model to simulate input data and assess rhapsodi's performance (*Figure 2—figure supplement 1*) while varying (a) the number of gametes, (b) the underlying number of hetSNPs, (c) depth of coverage, (d) recombination rate, and (e) genotyping error rate. Given these arguments, we construct a sparse matrix for input to the rhapsodi pipeline. The algorithmic steps of this generative model include (1) building phased diploid donor haplotypes (*Figure 2—figure supplement 1a*), (2) randomly tracing through these haplotypes to simulate recombination and construct gamete genotypes (*Figure 2—figure supplement 1b*), (3) masking genotype observations to mimic low sequencing coverage (*Figure 2—figure supplement 1c*), (4) superimposing genotype errors at a given rate (*Figure 2—figure supplement 1d*), and (5) filtering to only retain observable hetSNPs. We detail each step in turn below.

In the first step of the generative model, we construct phased haplotypes of the diploid donor by randomly generating a vector of 0s and 1s (with equal probabilities) at a length equal to the underlying number of hetSNPs. This binary vector is then inverted to generate the complementary haplotype (*Figure 2—figure supplement 1a*).

We then use these simulated homologs to generate genotypes for each gamete (*Figure 2—figure supplement 1b*). Specifically, we sample a count of recombination breakpoints from a Poisson distribution, where lambda reflects the average recombination rate. Each recombination breakpoint is randomly placed on the respective chromosome, and genotypes occurring after that breakpoint are inverted. In cases where a gamete chromosome does not recombine, the chromosome is identical to one of the parental chromosomes. By placing crossovers randomly, our method ignores heterogeneity in the landscape of meiotic recombination, as well as the phenomenon of crossover interference (*Broman and Weber, 2000*), which suppresses nearby crossovers.

After obtaining the gamete genotype matrix, we simulated the sparse coverage of single-cell sequencing data by masking observed genotypes. The missing genotype rate (MGR) is obtained from the probability density function of a Poisson distribution with $x=0$ and lambda equal to the sequencing coverage. We then multiplied the MGR by the total number of simulated SNP observations (number of underlying hetSNPs × number of gametes) to compute the total number of genotypes that should be masked (denoted as 'NA') (*Figure 2—figure supplement 1c*). Following a similar approach, we computed the number of genotyping errors by multiplying the error rate by the number of non-missing genotypes and then randomly placed these errors at individual SNPs by inverting the

respective alleles (replacing 0 with 1 or vice versa) (*Figure 2—figure supplement 1d*). Finally, SNPs or rows with at least one 0 genotype and one 1 genotype are kept (and the rest of the rows are filtered out), retaining only sites where both alleles were observed. In this way, the final number of observed hetSNPs in the generated output may be less than or equal to the specified number of underlying hetSNPs in the input.

We used this generative model to construct input data for ranges of gamete sample sizes (3, 15, 50, 150, 500, 1000, 2500, 5000), underlying numbers of hetSNPs (5000, 30,000, 100,000), and depths of coverage (0.001, 0.01, 0.1, 0.223, 0.357, 0.511, 0.693, 1.204, and 2.303×, corresponding to missing genotype rates of 99.9, 99, 90, 80, 70, 60, 50, 30, and 10%, respectively). For each combination of these three parameters, nine different conditions were simulated, with three independent runs for each condition. The conditions were defined by combinations of the genotyping error rate (0.001, 0.005, and 0.05) and recombination rate (0.6, 1, and 3) used to construct the gamete data. Simulations in which no hetSNPs remained after filtering were dropped from downstream benchmarking (2% of the generated datasets). In total, we produced 5713 simulations for downstream benchmarking across these conditions.

We evaluated the quality of donor haplotype phasing by considering the switch error rate, largest haplotype segment, completeness, and phasing accuracy (see Materials and methods section titled 'Definition of metrics'). To assess the quality of imputation of gamete genotypes, we quantified accuracy, completeness, and switch error rate. Finally, we assessed meiotic recombination discovery by computing the True Positive Rate (TPR), False Discovery Rate (FDR), and meiotic recombination breakpoint resolution. Given the imbalance in the number of true recombination breakpoints compared to the number of false recombination breakpoints, we also considered precision and F1 Score.

## Definition of metrics

- Accuracy (Phasing and Imputation): For two sequences of genotypes, s1 and s2 (of equal length and both represented by vectors comprising 0, 1, and NA), accuracy was defined as the Hamming error rate subtracted from 1. The Hamming error rate was defined to be equal to the number of positions with mismatches between s1 and s2 (ignoring a mismatch when a single sequence had an NA value) divided by the total number of sequence positions (*Li et al., 2020*; *Porubsky et al., 2017*).
- Completeness: For a sequence of genotypes, s1 (represented by 0 s, 1 s, and NAs), completeness was defined as the number of non-NA positions divided by the total number of positions in s1 (*Li et al., 2020*).
- Switch Error Rate: For two sequences of genotypes, s1 and s2 (of equal length and both represented by 0 s, 1 s, and NAs), the switch error rate was defined as the number of first mismatches in any stretch of adjacent mismatches (where the stretch is greater than or equal to length 1 and uninterrupted by NAs) divided by the total number of sequence positions (*Li et al., 2020*). This is visualized in *Figure 2—figure supplement 3*.
- Largest Haplotype Segment: For two sequences of genotypes, s1 and s2 (of equal length and both represented by 0s, 1s, and NAs), the largest haplotype segment was defined as the maximum number of adjacent matched positions, uninterrupted by NAs. For plotting purposes, we divide this number by the total number of sequence positions (*Li et al., 2020*).
- True Positive (TP): A true simulated meiotic recombination breakpoint intersects with predicted meiotic recombination breakpoint (*Figure 2—figure supplement 5b*).
- True Negative (TN): Simulated chromosome with no meiotic recombination events is correctly predicted to have no recombination breakpoints (*Figure 2—figure supplement 5b*).
- False Positive (FP): A predicted meiotic recombination breakpoint does not intersect with any true simulated meiotic recombination breakpoints (*Figure 2—figure supplement 5b*).
- False Negative (FN): A true simulated meiotic recombination breakpoint does not intersect with any predicted meiotic recombination breakpoints (*Figure 2—figure supplement 5b*).
- Recall or True Positive Rate (TPR): $\frac{TP}{TP+FN}$
- Precision: $\frac{TP}{TP+FP}$
- F1 Score: $\frac{2 \times precision \times recall}{precision+recall}$
- False Discovery Rate (FDR): $\frac{FP}{TP+FP}$
- False Positive Rate (FPR): $\frac{FP}{TN+FP}$
- Specificity or True Negative Rate (TNR): $\frac{TN}{TN+FP}$
- Accuracy (Discovery): $\frac{TP+TN}{TP+TN+FP+FN}$

## Automatic phasing window size calculation

We noted that for specific scenarios involving low coverages or small numbers of gametes, the positive relationship between phasing performance and increasing amounts of data (i.e., gamete sample size and coverage) was not always monotonic. This suggested that other parameters that were held fixed (notably, window size) may interact with sample size and coverage to influence performance.

To address this possibility, we performed a series of simulations (n=860) across a range of window sizes, tracking the phasing accuracy for each window size possibility. These simulations covered 459 unique combinations of number of gametes, number of SNPs, coverage, genotyping error, and recombination rate (1–3 independent replicate simulations each). For each set of parameters, the optimal window size which led to highest phasing accuracy was recorded. In cases of ties, the larger window sizes were preferred.

Optimal window size was expressed as a proportion (optimal number of SNPs per window / total number of SNPs on the simulated chromosome) such that data was bound between 0 and 1. We randomly designated 25% (n=213) of simulations as the test set, while using the remaining 75% (n=647) of simulations were designated as the training set. We fit a beta regression to the training data with the optimal window proportion as the response variable and number of gametes, coverage, genotyping error rate, and recombination rate as the predictor variables. As expected, we found that the number of gametes ($\hat{\beta} = 6.6 \times 10^{-4}$, p$-$value $= 2.2 \times 10^{-41}$), coverage ($\hat{\beta} = 2.1 \times 10^{-1}$, p-value $= 0.009$), and recombination rate ($\hat{\beta} = 3.8 \times 10^{-1}$, p$-$value $= 2.7 \times 10^{-18}$) were significant predictors of optimal window size. The model performed well on both the training and the held-out test set (*Figure 2—figure supplement 4*) with a training pseudo R-squared of 0.32 and no obvious loss of generalization in performance in the test set.

The results of this model are currently provided via an optional feature in the rhapsodi software package for automatic selection of phasing window size.

## Benchmarking run time

To estimate rhapsodi's runtime, we simulated sparse gamete datasets across the same range of coverage (0.001–2.303×) and gamete sample size (3–5000) as used in our broader benchmarking analysis (*Figure 2*). To be conservative, we used 100,000 hetSNPs for all simulations, equivalent to the largest dataset used in the other benchmarking analyses. Each simulation was run in triplicate with different random seeds. We then applied the `rhapsodi_autorun()` function to each dataset on 48 Intel(R) Xeon(R) Gold 6248 R CPUs @ 3.00 GHz. All simulations of 2500 gametes or fewer were run with an allocation of 189 GB of RAM. Simulations of 5000 gametes were run with an allocation of 1.47 TB of RAM. Time was recorded using R's `system.time()` function, and CPU seconds were calculated by summing together the user and self times for parent and child processes. All analyses took under 90 min of wall-clock time to run (*Figure 2—figure supplement 10*).

## Comparison to existing methods

We compared our method to the software package Hapi (haplotyping with imperfect genotype data) (*Li et al., 2020*), which was similarly developed for phasing and imputation based on single-cell DNA sequencing of gametes. Hapi was demonstrated to outperform the only other haploid-based algorithm, PHMM (pairwise Hidden Markov Model) (*Hou et al., 2013*), as well as two commonly applied diploid-based phasing methods, WhatsHap (*Martin et al., 2016*) and HapCUT2 (*Edge et al., 2017*) in terms of accuracy, reliability, completeness, and cost-effectiveness. Because Hapi outperforms PHMM and the PHMM software was not readily available, we did not directly compare rhapsodi to this method.

In contrast to rhapsodi, which is intended for datasets where many gametes are sequenced at individually low coverages, Hapi was designed for datasets that contain relatively small numbers of gametes (~3-7) sequenced at individually higher coverages (>1×). Given this distinction, the default behavior of Hapi's autorun function was not appropriate for our simulated datasets. In particular, Hapi's imputation function (`hapiImpute()`) by default filtered out any SNPs that had a missing genotype observation for at least one gamete, which results in all data being removed from our low-coverage datasets. We thus disabled this filtering behavior prior to benchmarking against our method to facilitate comparison.

As part of Hapi's autorun function, the HMM used to filter errors uses a fixed set of transition and emission probabilities. To more fairly compare the tools to each other, we instead provided Hapi with an HMM with parameters matching those of our simulation, thus closely mirroring the approach we used to evaluate rhapsodi in this analysis. After rhapsodi and Hapi were run on each dataset, the performances of both methods were evaluated using rhapsodi's assessment functions.

We then compared the default phasing method in rhapsodi (windowWardD2) to the modified HapCUT2 phasing approach (linkedSNPHapCUT2). Following the same steps outlined in Materials and methods section titled 'Assessing performance with simulation', we calculate the accuracy and completeness for phasing using each method and take the difference between the measures for each of the two methods *Figure 3—figure supplement 3*. The time requirements of linkedSNPHapCUT2 follow the same steps as those outlined in Materials and methods section titled 'Benchmarking run time', with the additional use of the Unix time function for the calling of HapCUT2. Evaluation of linkedSNPHapCUT2 does not include the timing of file conversion for use with HapCUT2, only the phasing; timing of windowWardD2 only evaluates the timing of phasing itself because no file conversion is necessary. We report both total user and system time as well as real time *Figure 3—figure supplement 4*. Simulations of linkedSNPHapCUT2 were allowed to time out if not completed after three days (including both file conversion and execution of HapCUT2's phasing).

While leveraging the data from *Bell et al., 2020*, rhapsodi offers substantive changes in the analysis approach. First, the Methods presented here build a reproducible software toolkit for similar analyses. Second, rhapsodi takes full advantage of the co-inheritance patterns of the SNP alleles; this phasing method is a principled approach tailored to the structure of the input data and knowledge of the biological process of meiosis. Third, for crossover discovery, rhapsodi assigns gamete-level genotypes via HMM-based imputation prior to detecting recombination breakpoints. Dealing with the error prior to crossover discovery enables rhapsodi to use the strengths of the HMMs.

## Inferring genetic similarity of sperm donors to reference samples

For inference of genetic similarity of donor individuals to reference samples from the 1000 Genomes Project (*Figure 5—figure supplement 2*; *Auton et al., 2015*), we merged data from all sperm cells of each donor, effectively mimicking sequencing of the donors' diploid genomes. We then randomly downsampled coverage 10-fold and limited analysis to chromosome 21 for computational tractability. Variants were called across all donor samples using freebayes (*Garrison and Marth, 2012*) with default settings and restricted to SNPs (as opposed to insertions or deletions). The list of variants observed in the Sperm-seq sample was then intersected and merged with genotype data from the 1000 Genomes Project reference panel of 3202 globally diverse samples (*Auton et al., 2015*). Using PLINK (*Purcell et al., 2007*), principal component analysis was applied to these merged genotype data, visualizing sperm donor sample principal component scores with respect to the labelled reference panel.

## Genotype filtering to mitigate spurious TD signatures

Using metadata from *Bell et al., 2020* and *Leung et al., 2021*, we removed any sperm cells designated as poor quality and any cell that was called as aneuploid for the chromosome of interest by the original researchers. To limit potential artifacts, we excluded all technically challenging regions of the genome previously identified by the Genome in a Bottle Consortium (GRCh38 'union') (*Krusche et al., 2019*) and/or the ENCODE Consortium (*Amemiya et al., 2019*). These include tandem repeats, homopolymers >6 bp, imperfect homopolymers >10 bp, difficult to map regions, segmental duplications, GC content <25% or >65%, and other problematic regions. To further filter individual variants, we restricted our analysis to SNPs that were also present in the 1000 Genomes Project dataset (*Auton et al., 2015*). Notably, even if TD were to be caused by a rare variant (which we would have removed from our dataset in this step), it is expected to occur on a haplotype that also carries common variants shared with the 1000 Genomes Project, allowing us to discern its effect. Finally, we removed any SNPs exhibiting an excess of observations across sperm cells (>1 standard deviation above the mean), with the goal of filtering out potential segmental duplications that are unique to a given donor. Such duplications contribute to alignment artifacts, whereby both copies of the sequence (including divergent sites) would pile up at the same location in the reference genome. This phenomenon may generate false signatures of TD, which we detail here.

Specifically, we identified two potential indicators that a given signature of TD may arise from rare segmental duplications (*Figure 5—figure supplement 3*). First, we consider the number of sperm in which the SNP is observed to carry the reference allele vs. the alternative allele (*Figure 5—figure supplement 3a*, *Figure 5—figure supplement 3b*). In the absence of TD, we expect points (each representing an SNP) to cluster to the line with slope = 1, indicating balanced representation of the reference and the alternative allele across the sample of sperm from that donor. Such is the case in *Figure 5—figure supplement 3b*, which is representative of most loci throughout the genome. However, if no filtering is applied, we observe that certain regions with strong TD signatures exhibit predictable patterns of allelic imbalance manifesting here as lines with slope = 0.5 and slope = 2 (*Figure 5—figure supplement 3a*). The second indicator that such regions derive from a rare segmental duplication regards the overall number of genotype observations (*Figure 5—figure supplement 3c*). Because of the low coverage (~0.01×), we expect most SNPs to be covered by one or more reads in only a few of the 969–3377 sperm per donor. If no filtering is applied, we find that SNPs exhibiting strong signatures of TD also exhibit an extreme excess of genotype observations, again consistent with duplication. To avoid these potential confounding impacts of segmental duplications, we set a stringent threshold for our filtering pipeline, removing SNPs with excess (>1 standard deviation above the mean) genotype observations across the pool of sperm. We calculated these means and standard deviations separately for each donor's chromosomes (25 donors × 22 chromosomes).

## Power analyses for detecting TD

To evaluate the statistical power of our TD scanning approach, we conducted simulations of varying levels of TD (0–20% deviations from Mendelian expectations) across a range of sample sizes of human sperm and applied two-tailed binomial tests (*Figure 4*). We used a Bonferroni correction to account for multiple hypothesis testing across both SNPs and donor individuals, as described in the Materials and methods section titled 'Significance threshold for TD scan'. Power is calculated here based on fully known gamete genotype data. While donors with fewer gametes may experience slightly decreased imputation and thus decreased power, even for the smallest sample size encountered in our study ($n$=969 sperm cells from a single donor), we reach both high accuracy and high completion in genotype imputation (*Figure 2b*). Larger samples increase our power to detect even smaller deviations from binomial expectations. Power was computed for each study design (number of gametes and rate of transmission) based on 1000 independent trials.

To evaluate our power to discover very strong TD, whereby the causal SNP and nearby SNPs may appear as homozygous across the sperm sample, we also applied rhapsodi to simulated data with a transmission rate ($k$) of 0.99. To simulate TD signatures, we developed a modified version of our generative model, whereby we randomly selected a causal SNP and haplotype to be under-transmitted relative to the other (*Figure 4—figure supplement 2*). We then generated a larger than desired set of gametes, removed a specified fraction of gametes containing this SNP/haplotype combination from the dataset, and sampled the desired number of gametes from this pool. By setting the fraction of gametes to remove, we effectively control the transmission rate ($k$), based on the following equation: $2 - (1/k)$.

Replicates of 100 independent simulations were performed across each chromosome for each of three donors (100 × 22 × 3 = 6600 total simulations): NC26 (average coverage, but small number of gametes), NC2 (slightly higher than average coverage and average number of gametes), and FF3 (lowest coverage, largest number of gametes). For each set of simulations, the coverage, number of gametes, and number of SNPs were set to match the data profile of that chromosome and donor combination, and datasets with strong TD ($k$ = 0.99) were simulated with the modified generative model. These simulated datasets were assessed with rhapsodi, and a TD scan was applied to the imputed data. The lead observed SNP was recorded and its p-value compared to the p-value threshold of $1.78 \times 10^{-7}$ used with the Sperm-seq TD scan. Power was then computed as the proportion of simulated lead observed hetSNPs with p-values below this significance threshold. The observed transmission rates for the lead observed hetSNPs ranged from 0.82 to 0.996 across the 6600 simulations.

## Genome-wide scan for TD

To scan the genome for TD, single-cell genotype calls were generated from sperm sequencing data as described by *Bell et al., 2020*. Briefly, genomic variants were called for each donor using all

sequence data from all sperm cells. After selection of heterozygous sites, the software GenotypeSperm (Drop-seq Tools v2.2, https://github.com/broadinstitute/Drop-seq/releases) was used to determine which sites were captured by each sperm cell barcode and which allele(s) were present at each of these observed sites. Aneuploidy status and cell doublets were identified as previously described (*Bell et al., 2020*; *Leung et al., 2021*). Only cell barcodes associated with single sperm cells (non-doublets) with even sequence coverage were included in the analysis, while aneuploid autosomes were excluded from the analysis.

We used a two-tailed binomial test (comparing counts of each allele under a null hypothesis of balanced transmission) to scan for TD at each hetSNP across sperm genomes from each of 25 donors. As input to this analysis, we used the imputed gamete data output by rhapsodi, applying the option to superimpose any observed genotypes in cases of discordance with inferred genotype state.

## Significance threshold for TD scan

As described above, we applied a separate binomial test for TD at each hetSNP in each donor individual. Given the relative infrequency of recombination (per base pair per meiosis), sperm genomes are composed of large tracts of the donor's two parental haplotypes. By consequence, genotype data from the sample of sperm from a given donor are affected by extreme LD, which typically extends across entire chromosomes. As such, our binomial tests are highly correlated across the genome, and naive multiple testing correction based on the number of SNPs would be overly conservative.

To account for this effect, we applied the method simpleM (*Gao et al., 2008*; *Gao et al., 2010*), which uses a PCA approach to compute the effective number of independent statistical tests. For the Sperm-seq dataset (combining across all donor individuals), simpleM estimated 281,368 effective tests (compared to 34,799,282 nominal tests). We then used the effective number of independent tests to apply a Bonferroni correction of $0.05/281{,}368 = 1.78 \times 10^{-7}$.

## Calculation of global signal of TD

We quantified global evidence of TD across the sperm genomes by calculating the rate of allele sharing between all pairs of sperm from each donor. First, we applied PLINK (*Purcell et al., 2007*) to the fully imputed sperm genotype matrix output by rhapsodi to generate a list of the SNPs deemed to segregate in near linkage equilibrium (`plink –file <infilename> –no-fid –no-parents – no-sex –no-pheno –indep 50 5 2 –out <outfilename>`). We then computed a Hamming distance matrix, effectively counting mismatches between each pair of sperm genomes from each donor. After repeating the above steps for all donors, we reported the mean across all pairs of sperm cells.

Null simulations were conducted by applying a modified version of the generative model (without missing genotypes or errors), using gamete numbers, pruned SNP counts, and crossover counts identical to those inferred from the empirical data. We repeated the above procedure for computing pairwise distances on each simulated dataset, recording the mean proportion of shared alleles across all sperm from each of the 25 donors. Repeating this entire procedure 500 times allowed the construction of a null distribution to which we compared our allele sharing calculation from the Sperm-seq data. We then computed a one-tail p-value, defined as the proportion of null simulations where the average pairwise allele sharing was greater than or equal to the observed value.

## Power analysis for detecting TD with pedigrees

Pedigree-based studies typically test for TD using the transmission disequilibrium test (TDT) (*Spielman et al., 1993*). The TDT is a McNemar's test of the binomial ($H_0$: $p_{A1} = p_{A2} = 1/2$), where $p_{A1}$ is the probability of transmitting the A1 alele and $p_{A2}$ is the probability of transmitting the A2 allele (*Meyer et al., 2012*). As a point of comparison for the single-sperm analysis, we computed the power of the pedigree-based McNemar test (using the test statistic $X = (b - c)^2/(b + c)$, where $b$ and $c$ are the numbers of observed transmissions of the A1 and A2 alleles, respectively *Meyer et al., 2012*) at varying levels of TD (0–20% deviations from Mendelian expectations) across a range of sample sizes of informative transmissions (*Figure 4—figure supplement 4*). The sample size refers to the number of offspring in which transmission of the allele of interest can be assessed (i.e., one heterozygous parent). We then plot the number of informative transmissions per SNP using published data from *Meyer et al., 2012* (their supplementary table 1).

## Availability of data and materials

Data analysis scripts specific to our study are available at https://github.com/mccoy-lab/transmission-distortion, (copy archived at swh:1:rev:d30bde24ca6d385036b88527c66a4a7d6261f8a5; *Carioscia et al., 2022*). Our package rhapsodi is available at: https://github.com/mccoy-lab/rhapsodi, (copy archived at swh:1:rev:8a5d712b1eb500594ac75428aa8dd94494bf81f3; *Weaver et al., 2022*), and a vignette detailing use of rhapsodi is included there, at https://github.com/mccoy-lab/rhapsodi/blob/master/vignettes/rhapsodi.html.

Raw sperm sequencing data from *Bell et al., 2020* can be accessed via dbGaP (study accession number phs001887.v1.p1), as described in the original publication. Raw sperm sequencing data from *Leung et al., 2021* was accessed upon request from the authors. We filtered the cells in our analysis using metadata published by *Bell et al., 2020* at: https://zenodo.org/record/3561081#.YLAdO2ZKhb9. Analogous metadata from *Leung et al., 2021* was obtained upon request from the authors.

## Acknowledgements

We thank members of the McCoy lab for feedback and helpful discussions. Thank you to Angela Leung and Alec Wysoker for assistance accessing Sperm-seq data. We also thank the staff at the Advanced Research Computing at Hopkins for computing support.

## Additional information

### Competing interests

Avery Davis Bell: is an inventor on a US Patent Application (US20210230667A1, applicant: President and Fellows of Harvard College) relating to the Sperm-seq single-cell sequencing method. Was an occasional consultant for Ohana Biosciences between October 2019 and March 2020. The other authors declare that no competing interests exist.

### Funding

| Funder | Grant reference number | Author |
| --- | --- | --- |
| National Science Foundation | 1746891 | Sara A Carioscia |
| National Institutes of Health | R35GM133747 | Rajiv C McCoy |

The funders had no role in study design, data collection and interpretation, or the decision to submit the work for publication.

### Author contributions

Sara A Carioscia, Software, Formal analysis, Funding acquisition, Investigation, Visualization, Methodology, Writing – original draft, Writing – review and editing; Kathryn J Weaver, Data curation, Software, Formal analysis, Validation, Investigation, Visualization, Methodology, Writing – original draft, Writing – review and editing; Andrew N Bortvin, Data curation, Software, Formal analysis, Investigation, Visualization, Methodology, Writing – original draft, Writing – review and editing; Hao Pan, Software, Formal analysis, Investigation, Methodology; Daniel Ariad, Methodology; Avery Davis Bell, Data curation, Investigation, Methodology, Writing – review and editing; Rajiv C McCoy, Conceptualization, Resources, Software, Supervision, Funding acquisition, Visualization, Methodology, Writing – original draft, Writing – review and editing

### Author ORCIDs

Sara A Carioscia http://orcid.org/0000-0002-0844-615X
Kathryn J Weaver http://orcid.org/0000-0001-8701-4197
Andrew N Bortvin http://orcid.org/0000-0001-8784-6786
Hao Pan http://orcid.org/0000-0002-5543-4792
Daniel Ariad http://orcid.org/0000-0002-7427-6435

Avery Davis Bell http://orcid.org/0000-0002-1837-302X
Rajiv C McCoy http://orcid.org/0000-0003-0615-146X

**Decision letter and Author response**
Decision letter https://doi.org/10.7554/eLife.76383.sa1
Author response https://doi.org/10.7554/eLife.76383.sa2

## Additional files

### Supplementary files
• Transparent reporting form

### Data availability

Data analysis scripts specific to our study are available at https://github.com/mccoy-lab/transmission-distortion (copy archived at swh:1:rev:d30bde24ca6d385036b88527c66a4a7d6261f8a5). Our package rhapsodi is available at: https://github.com/mccoy-lab/rhapsodi, (copy archived at swh:1:rev:8a5d712b1eb500594ac75428aa8dd94494bf81f3). Raw sperm sequencing data from Bell et al. (2020) can be accessed via dbGaP (study accession number phs001887.v1.p1), as described in the original publication. Raw sperm sequencing data from Leung et al. (2021) was accessed upon request from the authors. We filtered the cells in our analysis using metadata published by Bell et al. (2020) at: https://zenodo.org/record/3561081#.YLAdO2ZKhb9. Analogous metadata from Leung et al. (2021) was obtained upon request from the authors.

The following previously published dataset was used:

| Author(s) | Year | Dataset title | Dataset URL | Database and Identifier |
|---|---|---|---|---|
| Bell AD, Mello CJ, Nemesh J, Brumbaugh SA, Wysoker A, McCarroll SA | 2019 | Single-Sperm Genome Sequencing of Sperm Donors | https://www.ncbi.nlm.nih.gov/projects/gap/cgi-bin/study.cgi?study_id=phs001887.v1.p1 | dbGaP, phs001887.v1.p1 |

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
