## [Editor Report]

The paper reports a method to study deviations from Mendelian inheritance in genomic data from gametes. The authors use this method to study the existence of the phenomenon in human sperm data but do not find it. The method will be useful for future studies on segregation distortion, and the findings are an important step for the systematic study of segregation distortion in humans and other organisms.

---

## [Decision Letter]

**Decision letter after peer review:**

[Editors’ note: the authors submitted for reconsideration following the decision after peer review. What follows is the decision letter after the first round of review.]

Thank you for submitting the paper "Strict adherence to Mendel's First Law across a large sample of human sperm genomes" for consideration by *eLife*. Your article has been reviewed by 3 peer reviewers, one of whom is a member of our Board of Reviewing Editors, and the evaluation has been overseen by a Senior Editor. The reviewers have opted to remain anonymous.

Comments to the Authors:

Your manuscript was reviewed by three experts in the field and they concurred that the research is important and valuable. In particular, the development of the rhapsodi will be important for future studies. We also appreciated the application to human gamete data. The conclusions of the study largely confirm the results from previous studies and for that reason, we have decided to decline the manuscript. The work will not be considered further for publication by *eLife*. We hope the extensive comments from the reviewers will be useful.

*Reviewer #1 (Recommendations for the authors):*

The manuscript reports a new method (rhapsodi) to impute haplotypes in the sequencing of human gametes. The method performs well in simulated data and the authors explore parameter spaces relevant to the research. In particular, the authors evaluate how well the method performs at different genotyping depths and number of gametes. Even with relatively low coverage (~0.1X), the method phases the haplotypes effectively. Next, the authors evaluate the performance of rhapsodi to mis-specification of recombination and genotype error rate. Some simulations show a non-monotonic relationship with coverage which means the method is currently hard-coded for an unidentified source of variation. The authors acknowledge this caveat.

The second part of the manuscript compares the results from rhapsodi with those Hapi, another tool to phase haplotypes in gamete sequencing data and show that rhapsodi method seems superior in terms of computational time, completeness, and accuracy.

The next, and final, part of the manuscript is to analyze the data from the SpermSeq dataset (Bell et al. 2020; ~41K sperm haploid genomes). The authors do several steps of filtering and use the reduced dataset to infer crossover location along the genome (and the results are consistent with those of previous reports). Most importantly, the authors also use the method to infer Transmission Distortion (TD). There are a few sites (e.g., seven linked SNPs in chromosome 2) that show some deviations, but after using a multiple hit correction, there is no single SNP that shows a strong signal of TD. In general the manuscript concludes there is no significant deviations from random segregation along the whole human genome. This is a rigorous study that provides a novel tool for the study of gamete genomes. Nonetheless, the conclusion is not particularly novel. Perhaps if the authors provide more context on how the findings are contextualized in our current understanding of TD in humans, the manuscript would be more appealing. As it stands, the discussion is almost entirely about the method performance. An example of this lack of context comes in paragraphs 1 and 2 of the discussion. I would elaborate on the studies that are 'limited in statistical power'. Paragraph 2 suggest that previous studies were problematic but the authors do not describe the specifics. This segment of the discussion could be expanded for the benefit of the reader.

*Reviewer #2 (Recommendations for the authors):*

As mentioned briefly in the public review, the method developed in this work is well-motivated, high performing, and clearly explained. The descriptions of both simulations and application of the method to real data are very thorough, and the authors have taken care to reduce potential confounding effects of genotyping error and other data artifacts. The highlighted results regarding human TD are challenging to interpret in the context of various limitations, including an absence of discussion of how the method might perform under strong TD, as well as missing or unclear information about the TD simulations and the source of the samples. While some limitations of the TD analysis are discussed, other important ones are missing. Overall, the paper may be stronger if it focuses on the method rather than the TD analysis and/or incorporates or suggests an additional use case beyond searching for TD.

Major suggestions for improvement:

– Expand simulations of TD beyond 70% overtransmission of one allele, to identify an upper bound for TD beyond which allelic dropout substantially reduces power.

– The methods and Figure 4 – —figure supplement 4 provide only the number of gametes and not the sequencing coverage used for simulations of TD. Include information about coverage for these simulations, and highlight in the figure the range of parameters/conditions represented in the real dataset.

– The authors claim that they have ">80% power to detect even subtle TD." However, the results from the analysis with multiple test correction shown in Figure 4 – —figure supplement 4B show 80% power for a sample size of ~1000 gametes (the smallest sample size in the sperm dataset) at roughly a 0.58 transmission rate (it is challenging to see precisely from the colors in the figure). It would be helpful to provide an exact lower bound for the detection of TD under realistic conditions. A transmission rate of 0.58 is arguably not subtle and is comparable to the rate at which pedigree studies would also be reasonably well powered.

– In several places the authors state that a true positive signature of TD would not be eliminated through stringent filtering due to the persistence of strong LD. One effective way to demonstrate this would be to show that their filtering has not masked large regions of the genome in some donors. What are the longest stretches in which all SNPs have been eliminated through filtering?

– Repetitive regions, particularly centromeres and telomeres, are often involved in TD in other species, yet these would be filtered out as "challenging regions" in this analysis. Is it possible to infer how well these regions might be captured through LD with included SNPs? The section of the Discussion describing cases of TD that might not be captured in this analysis should include this limitation.

– Relatedly, if the marginal signal of TD on chromosome 2 shown in Figure 4 and described in brief on pp. 5 – 6 is at the end of the genotype-able portion of the chromosome, it could represent stronger TD involving telomeric repeats in LD with this region. This possibility could be excluded or discussed.

– It would be helpful for interpreting the results to include more details about the source of the sperm samples, particularly the donors' ancestry and whether the donors were known to have any fertility issues. This would provide useful context for the possibility of population-specific TD in the Discussion, as well as any comparisons to results of earlier studies using other methods.

– The Conclusion could be substantially strengthened by suggesting other potential uses for this method, including but not limited to detection of TD. Which samples would be most interesting to use for subsequent scans of TD? What other uses might this method have in non-TD-related genetic or evolutionary studies?

*Reviewer #3 (Recommendations for the authors):*

This method for phasing and imputing gamete genotypes requires substantially more benchmarking. Although accuracy is of interest and reasonably well evaluated through simulations, nothing is presented about the computational performance. In particular, given that much larger Sperm-seq datasets must already be in production, it is important to determine if this approach will scale sufficiently for those applications. Runtimes and memory requirements should be reported.

The code is reasonably well documented. However, I am not certain I could reproduce the haplotype phasing approach nor the HMM-based on the description provided in the text. The description should be expanded substantially possibly in a supplemental text section. For instance, in the state transition diagram in Figure 1C, as shown it indicates that the HMM does not transition back from error states. Is that right or am I misunderstanding?

The section, "Power analysis for detecting TD" states that the power across simulations is 80%. Does this account for a multiple testing correction? Both corrected and uncorrected values should be reported.

Two biological factors might be worth some additional clarification. First, what do we know about the ancestry of the donors? E.g., are some individuals admixed? Admixture might be expected to "unmask" cryptic distorter/suppressor pairs. Second, it should be clearly stated in the discussion that this study does not exclude the possibility of rare strong distorters at modest or low frequencies in human populations.

To me, figure 4-S3 is somewhat misleading. Note that this might actually be figure 4-s4 given the caption and references in the text, but on the pdf that page is labelled figure 4-s3. None of the donors considered had more than about 3300 total sperm sampled. After QC, I assume it is less. Yet the X-axis proceeds to 10k. I'd like to see the X-axis pruned back to a reasonable range for these analyses and a line plotted with points for each donor where the experiment would have power to detect TD. I suggest plotting the line at 80% power by convention, but 50% would allow for a more direct comparison to Meyer et al. (2012). This really should be a main text figure, too, since all of the major biological claims in the paper hinge on these analyses it is necessary to explore power in much greater depth.

[Editors’ note: further revisions were suggested prior to acceptance, as described below.]

Thank you for resubmitting your work entitled "Adherence to Mendel's First Law across a large sample of human sperm genomes" for further consideration by *eLife*. Your revised article has been evaluated by Molly Przeworski (Senior Editor) and a Reviewing Editor.

The manuscript was reviewed by two of the original reviewers and two additional ones. Overall, the reviewers agree on the merit of the work. The manuscript has been improved but there are some important remaining issues that must be addressed. In summary, we have three requests and one suggestion.

Essential:

– A comparison to hapcut2 seems to be crucial for benchmarking the method. Since your method has a phasing component, it is important to present a comparison that is adequate given the scale of the dataset.

– The criticisms about pedigree-based studies need to be presented in a more nuanced way. Similarly, the broad statement about the absence of strong TD in humans seems poorly supported. The reviewers all suggested a more balanced presentation of these previous efforts and more of a discussion of how rhapsodi can be integrated in current research.

– The manuscript reads as two disjointed pieces, one on method development and the other on applications. We would ask that the manuscript be revised with this issue in mind, as it persists from the previous submission.

Potentially useful but not essential:

The reviewers list some suggestions, including changing the title, modifying the abstract, and reordering the introduction and discussion.

*Reviewer #2 (Recommendations for the authors):*

I appreciate the extensive further simulations, analyses, and revisions the authors have provided, particularly the additional simulations under strong TD and the clarified details about their samples, metrics, and computing resources. The suggestions for future uses of rhapsodi are also helpful in communicating the significance of this work. These additions substantially strengthen the manuscript and alleviate many of my initial concerns.

I still feel that statements about generalizing a negative finding in the search for TD within this sample to a broader statement about the absence of strong TD in humans (e.g., line 107 "underscores the fidelity of human male meiosis for ensuring balanced transmission of alleles to the gamete pool"), are somewhat too strong, for the following reasons:

1) All power simulations for rhapsodi-based TD detection are conditional on observing a distorter in heterozygous state within the sample, which would require the distorter not to be very rare (roughly >1.4% allele frequency required for 50% probability that at least one of 25 individuals is heterozygous). Observed distortion alleles in other species are often at very low frequency under an equilibrium model where the distortion is balanced by fitness costs in homozygotes (e.g., SD is found at 1 – 5% frequency per ref #9). Unbalanced distorters would fix or be lost rapidly, and would be unlikely to be detected except in the very brief window at which they are locally at intermediate frequency. The authors note these as caveats in the Discussion, but the need for a relatively common distorter is not described as an issue of power (see discussion of comparison to pedigree-based studies below).

2) I may have missed it, but I did not see mention of the possibility of population-specific TD aside from the discussion about fixed distorters uncovered through admixture. While it is true that there are very few alleles with extreme patterns of frequency differentiation across human populations (lines 620 – 621), this is not highly relevant for whether one should expect distorters to be population-specific. As mentioned above, distortion loci are likely to be rare and/or ephemeral, and either would make them very likely to be population-specific.

3) Some TD systems involve epistasis between alleles at distorter and responder loci, which would further require the responder allele to be observed heterozygous in the sample.

I did not notice this before, but there are statements in both the Introduction (line 63) and Discussion (line 505) about pedigree-based TD scans being underpowered due to the small size of human families. This is a bit misleading because such studies typically combine data across multiple families (as noted by the authors in lines 523 – 524), which reduces their power to detect TD present in any one individual but not to detect TD involving an allele at intermediate frequency that is heterozygous in many parents within the sample. Both pedigree-based and sperm sequencing studies have issues of power due to sample size of individuals that are currently discussed primarily in the context of pedigree-based studies; donor sample size concerns are separated from considerations of power when discussing the sperm sequencing study design.

The statement that "single-gamete sequencing studies… provide equal power for detecting TD involving common and rare alleles" is undermined by the following clause "provided that they are heterozygous in the sampled donor individual." The probability that any of the sampled donor individuals are heterozygous for a distorter depends upon that distorter's frequency in the population, so the study's overall power to detect TD is not equal for common and rare alleles (see point 1 above). For example, in the context of Figure 4—figure supplement 4, 200 informative transmissions out of 1518 trios would imply p(heterozygous) = 0.066 (200/3036, MAF = 0.034). In the Sperm-seq sample of 25 individuals, 18% of such cases would be expected to have no heterozygotes in the sample. As this example demonstrates, the power for the present study is substantially higher than for previous pedigree-based studies, but not as much higher as implied by the power simulations.

It might be worth considering for future work whether power may be gained for detecting TD using rhapsodi by combining data across individuals following haplotype inference.

*Reviewer #3 (Recommendations for the authors):*

Overall, the manuscript has improved and many of my concerns have been well addressed. In particular, emphasizing the power considerations of this specific approach in the main text was an important addition. I appreciate the authors' correcting inaccuracies in my understanding of their previous work.

I respectfully disagree with the authors' dismissal of several considerations.

First, the paper remains a pretty sharp subdivision of methods vs biology. The methods are really about phasing and recombination detection in sperm-seq data, the biology is about TD. I believe it will be hard for a non-specialist to read this manuscript, though the authors are correct that extremely specialized users --- i.e., those with their own sperm-seq datasets --- may benefit from improved software usability.

Second, a comparison to an approach that is suited to the scale of data used is necessary. If hapcut2 is the only option, it should be applied despite being an "off-label" use. Also, yes, it is somewhat outside of typical hapcut2, but linking the reads bioinformatically is pretty straightforward and reasonable. It bears some similarity to read-backed phasing Certainly if one of this studies' co-authors believed this to be a valid use in previous published work it is reasonable to include as a point of comparison here.

On a related note, in "Benchmarking against existing methods", while the description of previous analysis is accurate, the relevant underlying datasets vary so much from previous works that is it not clear that the same results would hold here. Minimally, some acknowledgement should be added that the results described from previous work are based on a dramatically different dataset than was considered here.

*Reviewer #4 (Recommendations for the authors):*

The authors describe two main things in this paper:

First is the software package called "rhapsodi". Rhapsodi is an R package which is designed to use sparse whole-genome sequencing data from gametes to phase a sperm donor's haplotypes, impute gamete genotypes and identify meiotic crossover breakpoints.

Second, rhapsodi was used with both simulated data and a published dataset of 41,189 individual sperm cells' sequence. The main biological focus of this second part of the paper involved testing for transmission distortion, of any alleles or broader linkage blocks, for which none is detected. Previous publications, referenced in the manuscript, have suggested that transmission distortion can occur in human populations. The possible causes of the discrepancies in results between the study and previous studies is discussed.

Strengths

In the set of variables assessed, rhapsodi appears to outperform Hapi, which the authors present as the benchmark to meet or exceed for a package focused on efficient, accurate, complete and reliable phasing of haplotypes.

Another strength is that the paper also assesses transmission distortion with particularly large datasets for which no signal is detected. This contrasts from previous studies cited in the manuscript. However, the claims relevant to the analysis of this public dataset appears to be convincing. Further, the authors go on to offer good reasoning, particularly in the Discussion, about how they arrived at these conclusions, what any potential limitations of the approach may be, and how the dataset used differs fundamentally from pedigree-based data i.e. this study investigates transmission rates in gametes and does not consider different stages of fertilisation and subsequent aspects embryonic development.

Weaknesses

The approaches in the paper generally are quite strong. However, the study lacks a positive control for transmission distortion, which is not simulated data. This positive control does not exist in published human datasets to the best of my knowledge. However, outside of the scope of this study, one could consider reanalysing published non-human single-gamete sequencing datasets of which there are a number of studies.

The analyses of meiotic crossovers with non-simulated data are quite light – limited to Figure 5 sup 4 and 5 – compared to the two other features of rhapsodi (phasing and imputation), which feature more extensively. Given that a significant portion of the manuscript presents a re-analysis of the data from Bell et al. 2019 Nature, it be helpful to visually demonstrate the variation in crossover numbers, positioning, interference between the 25 donors, and where possible compare these results to published analyses e.g. Bell et al. and other studies. For example, Figure 5—figure supplement 5 suggests that rhapsodi is much more conservative with calling of crossovers/haplotype transitions than Bjarni et al. and I think that this is worthy of discussion. Particularly because false positive rates of double crossovers can have a large effect on particular study types, such as those concerning crossover interference.

The outputs of this work will be very useful for future researchers; both the rhapsodi software, and the finding that there is no strong transmission distortion signal in these 25 male sperm samples. Rhapsodi is one of a very limited number of user-friendly generalised pieces of software – as opposed to a collection of scripts – for the analysis of sparsely sequenced gametes for haplotype phasing, imputation and meiotic crossover breakpoint analysis. The package should attract significant attention from the potential userbase.

The biological findings of this study – failure to identify transmission distortion in these samples – should be of general interest to geneticists and adjacent fields. Despite being a negative finding, it challenges existing literature with a robust analysis and will be sure to stimulate further research directions.

I found the paper very interesting. I also found the paper to be very well written, clear, accessible to a broad readership, and look forward to using rhapsodi.

*Reviewer #5 (Recommendations for the authors):*

The authors introduce a new method rhapsodi, which accurately infers haplotypes from low-coverage sequence data of large sample sizes of haploid gametes. They demonstrate that rhapsodi performs better than the existing approach Hapi, particularly for larger samples of gametes. They apply rhapsodi to test for evidence of transmission distortion (TD) in a published human Sperm-seq dataset (single cell sequencing of >30k sperm from 25 donors), and report no significant evidence of TD.

rhapsodi is a powerful approach, which will be useful in a variety of contexts. The TD analysis is more rigorous than previous pedigree-based studies, providing convincing evidence for a lack of strong distorters. However, power to detect weak TD remains limited after accounting for multiple testing. The method is a larger contribution, and likely to be of interest to a broader audience. The manuscript would benefit from re-framing to focus on introducing rhapsodi – with human TD results demonstrating its application.

Recommendations for authors:

The authors have thoroughly and carefully addressed the technical concerns of previous reviewers, and better placed the results in context with previous literature.

However, as noted by previous reviewers, the major strength of the manuscript is the method rather than the human TD analysis. Re-framing the manuscript with focus first on the method would better reflect the relative contribution of the aims, and broaden readership. This could be accomplished without extensive re-writing. For example, I suggest:

– Change title: The title refers only to the TD results – including mention of the method in the title will be more accurate and make it less likely to be overlooked by readers interested in applying the method to other questions. e.g. something like:

New method for analyzing haploid gamete sequences applied to a large sample of human sperm shows adherence to Mendel's first law.

– Re-ordering abstract: to first highlight the need for new method/software to analyse SpermSeq data – ie low-coverage sequencing data from large sample size of gametes. Next highlight TD in humans as an example of a question that can be addressed with SpermSeq data better than previous. [summary of findings] Add at the end more explicit mentions of other applications of rhapsodi (outlined in L653-672 of Discussion)

– Re-order Introduction/beginning of Discussion in similar manner.

Other comments:

L402 "even slight deviations from Mendelian expectations, as supported by our simulations of TD across a range of gamete sample sizes and transmission rates"

– I would consider ~50-55% slight deviations – power is low in this range

– Single noted peak on Chr 2 is 56% for 1571 gametes – but doesn't reach genome-wide significance

– This concern has been addressed somewhat by removing "strict" etc in response to previous reviewers, but could

L491 comparison to Zollner – excess of allele sharing among sibs in pedigree – 50.43% -

this is surviving offspring not gametes so reflects additional sources of TD and selection on embryos/offspring

L621-631 discusses other sources of TD but does not mention this may explain the discrepancy with pedigree based approaches

Could add within-ejaculate sperm competition as factor – reviewed in:

https://royalsocietypublishing.org/doi/full/10.1098/rstb.2020.0066#d1e627

L581 this paragraph makes more sense combined with points in L543-550

Filtering of repeat regions and segmental duplications could remove sites with TD (as noted by the previous reviewer) – in L545 point out the benefits of stringent filtering but doesn't point out that some of these regions could be effected by TD

Methods – in beginning, briefly reiterate the differences compared to methods in the sperm-seq paper (Bell et al. 2020) – it's unclear that the method here is similar but formalised into the package

Figure 4 – Figure supp 1 and Figure 2-supp9 – can't distinguish different colors – make points bigger and/or change shape also according to scale

---

## [Author Response]

[Editors’ note: The authors appealed the original decision. What follows is the authors’ response to the first round of review.]

Reviewer #1 (Recommendations for the authors):The manuscript reports a new method (rhapsodi) to impute haplotypes in the sequencing of human gametes. The method performs well in simulated data and the authors explore parameter spaces relevant to the research. In particular, the authors evaluate how well the method performs at different genotyping depths and number of gametes. Even with relatively low coverage (~0.1X), the method phases the haplotypes effectively. Next, the authors evaluate the performance of rhapsodi to mis-specification of recombination and genotype error rate. Some simulations show a non-monotonic relationship with coverage which means the method is currently hard-coded for an unidentified source of variation. The authors acknowledge this caveat.

We noted in the previous manuscript draft that for specific scenarios involving low coverages or small numbers of gametes, the positive relationship between phasing performance and increasing amounts of data (i.e., gamete sample size and coverage) was not always monotonic. This suggested that other parameters that we currently hold fixed (notably, window size) may interact with sample size and coverage to influence performance.

To address this possibility, we performed a series of simulations (n = 860) across a range of window sizes, tracking the resultant phasing accuracy. The optimal window size (maximum accuracy) was then identified for each simulation. We designated 25% (213) of these simulations as the test set, while using the remaining 75% (647) of simulations as a training set. For the training set, we fit a β regression with the optimal window size (number of SNPs in the window / total number of SNPs) as the response variable and number of gametes, coverage, genotyping error rate, and recombination rate as the predictor variables. As expected, we found that the number of gametes, coverage, and recombination rate were significant predictors of optimal window size. Additionally, the model performed well on both the training and the held-out test sets (Figure 2—figure supplement 10).

We have used the model above as the basis of an automatic window size calculation. We currently implement it as an optional feature within the software package and briefly describe these results in the manuscript text (Figure 2—figure supplement 10; Methods lines 988-1028). We plan to further develop this feature through more extensive simulations in the future.

The second part of the manuscript compares the results from rhapsodi with those Hapi, another tool to phase haplotypes in gamete sequencing data and show that rhapsodi method seems superior in terms of computational time, completeness, and accuracy.

In response to comments from another reviewer, we expanded the comparisons to Hapi in regard to computation time, while also providing additional context about why Hapi is the most relevant software for comparison (lines 287-295; lines 299-303).

The next, and final, part of the manuscript is to analyze the data from the SpermSeq dataset (Bell et al. 2020; ~41K sperm haploid genomes). The authors do several steps of filtering and use the reduced dataset to infer crossover location along the genome (and the results are consistent with those of previous reports). Most importantly, the authors also use the method to infer Transmission Distortion (TD). There are a few sites (e.g., seven linked SNPs in chromosome 2) that show some deviations, but after using a multiple hit correction, there is no single SNP that shows a strong signal of TD. In general the manuscript concludes there is no significant deviations from random segregation along the whole human genome. This is a rigorous study that provides a novel tool for the study of gamete genomes. Nonetheless, the conclusion is not particularly novel. Perhaps if the authors provide more context on how the findings are contextualized in our current understanding of TD in humans, the manuscript would be more appealing. As it stands, the discussion is almost entirely about the method performance. An example of this lack of context comes in paragraphs 1 and 2 of the discussion. I would elaborate on the studies that are 'limited in statistical power'. Paragraph 2 suggest that previous studies were problematic but the authors do not describe the specifics. This segment of the discussion could be expanded for the benefit of the reader.

Thank you for your careful reading and helpful suggestions. We agree that providing additional context about current understanding of TD in humans would improve our work. To this end, we bolstered both the Introduction and Discussion for a more thorough background about the strengths and limitations of pedigree-based studies, pooled gamete sequencing studies, and single-gamete sequencing studies in both humans and non-human systems (lines 37-46; lines 508-529; lines 557-580).

For additional context about why single-gamete sequencing is a powerful method, we explain that patterns of allelic co-inheritance among single sperm cells from the same donor individuals facilitate the imputation of missing genotype data even at very low depths of coverage (~0.01x) per cell, thereby augmenting our sample size at each heterozygous SNP by approximately 100-fold and improving statistical power compared to naive scans of aligned reads (lines 551-557).

In response to this and other reviewers’ comments, we have also extended our power analyses and more directly compared to power achieved by previous pedigree-based scans (Figure 4, Figure 4—figure supplement 1, Figure 4—figure supplement 4). Importantly, it is not the statistical test or pedigree nature of the data per se that limit statistical power, but rather the fact that the number of informative transmissions (i.e., the relevant sample size) per SNP in a pedigree based study is limited by the number of heterozygous individuals. This contrasts with our single-sperm analysis, where the relevant sample size for all observed SNPs is equal to the total number of gametes (lines 508-521).

Reviewer #2 (Recommendations for the authors):As mentioned briefly in the public review, the method developed in this work is well-motivated, high performing, and clearly explained. The descriptions of both simulations and application of the method to real data are very thorough, and the authors have taken care to reduce potential confounding effects of genotyping error and other data artifacts. The highlighted results regarding human TD are challenging to interpret in the context of various limitations, including an absence of discussion of how the method might perform under strong TD, as well as missing or unclear information about the TD simulations and the source of the samples. While some limitations of the TD analysis are discussed, other important ones are missing. Overall, the paper may be stronger if it focuses on the method rather than the TD analysis and/or incorporates or suggests an additional use case beyond searching for TD.

Thank you for these comments and suggestions, which have strengthened our manuscript.

Major suggestions for improvement:– Expand simulations of TD beyond 70% overtransmission of one allele, to identify an upper bound for TD beyond which allelic dropout substantially reduces power.

We added simulations of very strong TD (k = 0.99; Figure 4—figure supplement 3) as noted in the responses above. Our simulations support the accuracy of phasing and imputation under these circumstances. Importantly, while extreme TD does indeed result in “allelic dropout”, whereby the causal SNP and nearby flanking SNPs appear as homozygous (and thus undetectable without external knowledge of donor heterozygous SNPs), recombination recovers heterozygosity as one extends out in either direction along the chromosome, and massive signatures of TD become apparent within these regions (Power = 100% with coverage and gamete sample sizes resembling the Sperm-seq data).

Notably, while tracts of homozygosity do occasionally appear in the Sperm-seq data by consequence of data filtering and phenomena such as heterozygous deletions or copy neutral runs of homozygosity, no TD signal is apparent on the borders of these tracts as would be expected of strong TD (lines 471-485).

– The methods and Figure 4 – —figure supplement 4 provide only the number of gametes and not the sequencing coverage used for simulations of TD. Include information about coverage for these simulations, and highlight in the figure the range of parameters/conditions represented in the real dataset.

This point is related to points raised by Reviewer 3. To address these comments, we have modified this figure (now main Figure 4) to highlight the range of parameters (including coverage) that is consistent with the Sperm-seq dataset. These changes clarify the statistical power of our study and facilitate more direct comparison to power of previous studies from the literature (Figure 4, Figure 4—figure supplement 1).

– The authors claim that they have ">80% power to detect even subtle TD." However, the results from the analysis with multiple test correction shown in Figure 4 – —figure supplement 4B show 80% power for a sample size of ~1000 gametes (the smallest sample size in the sperm dataset) at roughly a 0.58 transmission rate (it is challenging to see precisely from the colors in the figure). It would be helpful to provide an exact lower bound for the detection of TD under realistic conditions. A transmission rate of 0.58 is arguably not subtle and is comparable to the rate at which pedigree studies would also be reasonably well powered.

We have added an additional figure (Figure 4—figure supplement 1) that more clearly illustrates our power to detect TD within the Sperm-seq dataset, as well as an additional figure that depicts the power of pedigree-based studies (Figure 4—figure supplement 4).

Our simulations of strong TD (Figure 4—figure supplement 3) and modification of the title (to remove the word “strict”) also help address this point, as does the additional context about pedigree, pooled gamete sequencing, and single gamete sequencing studies provided in the updated Introduction and Discussion (lines 37-46; lines 508-529; lines 557-580).

– In several places the authors state that a true positive signature of TD would not be eliminated through stringent filtering due to the persistence of strong LD. One effective way to demonstrate this would be to show that their filtering has not masked large regions of the genome in some donors. What are the longest stretches in which all SNPs have been eliminated through filtering?

We have noted in “Results: Adherence to Mendelian expectations across sperm genomes” that the longest observed tracts of homozygosity after filtering were ~3.5 Mbp (lines 476-485). Our TD simulations, however, demonstrate that given the extent of LD (which spans entire chromosomes) within a sample of sperm from a single donor, TD signal should still be apparent on the flanks of such regions where heterozygosity is restored (Figure 4—figure supplement 3). We have also added text to the Discussion further explaining why stringent filtering is necessary to avoid spurious signatures of TD (lines 541-550; lines 581-586)—a point that is not unique to our study, but that we approached very conservatively.

– Repetitive regions, particularly centromeres and telomeres, are often involved in TD in other species, yet these would be filtered out as "challenging regions" in this analysis. Is it possible to infer how well these regions might be captured through LD with included SNPs? The section of the Discussion describing cases of TD that might not be captured in this analysis should include this limitation.

We have updated the Discussion to acknowledge that a caveat of this and most previous scans for TD in humans is the inability to analyze highly repetitive and other technically challenging regions of the genome, which include heterochromatic sequences such as telomeres and centromeres that have been implicated in TD in other species (e.g., Brand et al. 2015, Axelsson et al. 2010). We also briefly describe resources and future technologies that could facilitate the analysis of TD within these technically challenging regions (lines 593-598).

– Relatedly, if the marginal signal of TD on chromosome 2 shown in Figure 4 and described in brief on pp. 5 – 6 is at the end of the genotype-able portion of the chromosome, it could represent stronger TD involving telomeric repeats in LD with this region. This possibility could be excluded or discussed.

In relation to the previous point, we now address the possibility of TD occurring at sites in LD with the region at the end of chromosome 2 in the Discussion (lines 586-593). We note that the stringent filtering of repetitive regions was necessary for avoiding spurious signatures of TD that may arise due to read mapping and genotyping artifacts, and we discuss future technologies that could facilitate genotyping within these regions.

– It would be helpful for interpreting the results to include more details about the source of the sperm samples, particularly the donors' ancestry and whether the donors were known to have any fertility issues. This would provide useful context for the possibility of population-specific TD in the Discussion, as well as any comparisons to results of earlier studies using other methods.

Using a principal component analysis (PCA) and data from the 1000 Genomes Project, we compared the genetic similarity of each donor relative to a reference panel. We have added this comparison as Figure 5—figure supplement 2. A total of 16 sperm donor individuals clustered with reference samples from the European superpopulation, 3 donors clustered with reference samples from the East Asian superpopulation, 2 donors clustered with reference samples from the South Asian superpopulation, and 3 samples clustered on an axis between the African and European superpopulations, consistent with their previously reported admixed African American backgrounds. Meanwhile, one sample (NC26) showed similarities with both the African and East Asian populations, again consistent with the previous report (lines 339-354).

We added information about the fertility status of each donor to the “Results: Application to data from human sperm” (lines 333-339). Of the 25 sperm donors, samples from 20 individuals were obtained from a sperm bank and of presumed (but unknown) normal fertility status (Bell et al. 2020), while 5 donors had known reproductive issues: failed fertilization after intracytoplasmic sperm injection (n=2) or poor blastocyst formation (n=3) (Leung et al. 2021).

In the Discussion (lines 613-621), we briefly consider the possibility of population-specific TD (a comment also raised by Reviewer 3). One related possibility raised by that reviewer is that admixture between populations could separate a (locally fixed) distorter from a (locally fixed) suppressor, revealing hidden distorter alleles. We hypothesize that this scenario is unlikely in humans, given that very few alleles exhibit such extreme patterns of frequency differentiation across human populations (1000 Genomes Project Consortium, 2015).

– The Conclusion could be substantially strengthened by suggesting other potential uses for this method, including but not limited to detection of TD. Which samples would be most interesting to use for subsequent scans of TD? What other uses might this method have in non-TD-related genetic or evolutionary studies?

Thank you for these suggestions, which we agree have strengthened our paper and established a starting point for others in the field to use our method.

As noted in our response to the previous point, the recent assembly of a complete human reference genome has resolved many heterochromatic sequences for the first time and offers great promise for the accurate study of TD genome-wide, especially with the use of long-read sequencing (Nurk et al. 2022, Aganezov et al. 2022).

In the Conclusion, we now also consider potential uses for rhapsodi in other genetic and evolutionary studies, beyond analysis of TD (lines 653-672). These include the construction of genetic maps for non-model species to complement other genomic resources such as genome assemblies. Within humans, the construction of personalized recombination maps could be useful to better understand the sources of variation in recombination and relation to phenotypes such as fertility status (Lyu et al. 2021). Notably, whole and partial chromosome gains and losses (i.e., aneuploidies) are the leading cause of human pregnancy loss, and one potential mechanism of aneuploidy formation is the abnormal number or placement of meiotic crossovers (Lamb et al. 2005). The chromosome-length haplotypes generated by rhapsodi offer additional context for population genetic analyses, including inferring kinship and population structure, detecting signatures of positive selection, and assessing demographic history (Li et al. 2020).

Reviewer #3 (Recommendations for the authors):This method for phasing and imputing gamete genotypes requires substantially more benchmarking. Although accuracy is of interest and reasonably well evaluated through simulations, nothing is presented about the computational performance. In particular, given that much larger Sperm-seq datasets must already be in production, it is important to determine if this approach will scale sufficiently for those applications. Runtimes and memory requirements should be reported.

We have added information in the Results regarding rhapsodi’s runtimes and memory requirements for processing datasets ranging from the size (number of gametes, number of SNPs, coverage) of the sperm data used here to datasets 10x larger. We detail these requirements in Figure 2—figure supplement 9.

In the section titled “Results: Evaluating performance on simulated data”, we show that while rhapsodi was rigorously benchmarked for datasets containing up to 5000 gametes with 100,000 SNPs at coverages up to 2.3x coverage, the method is capable of analyzing much larger datasets (assuming coverages and SNP densities comparable to the data in Bell et al. [2020]; 0.01x coverage, 90,795 SNPs). To demonstrate this point, we successfully applied rhapsodi to simulated datasets of up to 32,900 gametes in under 24 hours running multi-threaded on 48 CPU cores (lines 1029-1047). This represents a dataset 20x the size of that produced by Bell et al. (2020).

We reference these results and their implications for future studies in the Discussion. In summary, our simulations demonstrate that rhapsodi outperforms existing methods for phasing and imputation of single-gamete data, while also efficiently scaling to datasets 20-fold the size of the data analyzed in our study (lines 274-286).

The code is reasonably well documented. However, I am not certain I could reproduce the haplotype phasing approach nor the HMM-based on the description provided in the text. The description should be expanded substantially possibly in a supplemental text section. For instance, in the state transition diagram in Figure 1C, as shown it indicates that the HMM does not transition back from error states. Is that right or am I misunderstanding?

We added additional detail regarding the haplotype phasing approach in the section titled “Methods: Reconstruction of phased donor haplotypes” (lines 718-761). We have also provided additional detail regarding the gamete imputation approach and the HMM in the section titled “Methods: Imputation of gamete genotypes” (lines 762-831).

We also linked a vignette for using rhapsodi in the main text section “Availability of data and materials” (lines 1321-1340), which details each of the three algorithmic steps of rhapsodi.

The section, "Power analysis for detecting TD" states that the power across simulations is 80%. Does this account for a multiple testing correction? Both corrected and uncorrected values should be reported.

Thank you for raising this point. We have now clarified that we report Power without (Figure 4A) and with (Figure 4B) multiple testing correction. We similarly report both values in Figure 4—figure supplement 1.

Two biological factors might be worth some additional clarification. First, what do we know about the ancestry of the donors? E.g., are some individuals admixed? Admixture might be expected to "unmask" cryptic distorter/suppressor pairs. Second, it should be clearly stated in the discussion that this study does not exclude the possibility of rare strong distorters at modest or low frequencies in human populations.

In response to this comment and to Reviewer 2’s comment regarding the ancestry of the donor individuals, we compared the genetic relatedness of each donor using a principal component analysis (PCA) with human genetic data from the 1000 Genomes Project (Figure 5—figure supplement 2; lines 339-354). A total of 16 sperm donor individuals clustered with reference samples from the European superpopulation, 3 donors clustered with reference samples from the East Asian superpopulation, 2 donors clustered with reference samples from the South Asian superpopulation, and 3 samples clustered on an axis between the African and European superpopulations, consistent with their previously reported admixed African American backgrounds. Meanwhile, one sample (NC26) showed similarities with both the African and East Asian populations, again consistent with the previous report.

The potential for admixture to unmask distorter/suppressor pairs is a fascinating one that we now briefly address in the Discussion (lines 608-621). After distorter alleles arise, suppressors may subsequently evolve to re-balance the transmission ratio (e.g., Greenberg et al. 2020). Admixture between populations may then separate the distorter from the suppressor, revealing hidden distorter alleles. We hypothesize that this scenario is unlikely in humans, given that very few alleles exhibit such extreme patterns of frequency differentiation across human populations (1000 Genomes Project Consortium, 2015).

As you note, there is the potential for a rare, strong distorter to affect TD in only a subset of the population. Such distorter alleles may not be detected in our study, especially due to the small sample size. We agree with this point and discuss this fundamental limitation in the Discussion (lines 605-608).

To me, figure 4-S3 is somewhat misleading. Note that this might actually be figure 4-s4 given the caption and references in the text, but on the pdf that page is labelled figure 4-s3. None of the donors considered had more than about 3300 total sperm sampled. After QC, I assume it is less. Yet the X-axis proceeds to 10k. I'd like to see the X-axis pruned back to a reasonable range for these analyses and a line plotted with points for each donor where the experiment would have power to detect TD. I suggest plotting the line at 80% power by convention, but 50% would allow for a more direct comparison to Meyer et al. (2012). This really should be a main text figure, too, since all of the major biological claims in the paper hinge on these analyses it is necessary to explore power in much greater depth.

Thank you for your careful consideration of our power analysis figure, which is a crucial aspect of our manuscript. We agree with your feedback and have improved the figure to address your comments.

Specifically, as suggested, we have made this a main text figure (Figure 4) with the X-axis pruned back to a reasonable range for these analyses: 500-3500 gametes for a given individual, reflective of our study design. We also added additional supplementary panels (Figure 4—figure supplement 1) that more clearly show the power to detect various strengths of transmission distortion across the sample sizes used in our study, including 80% power as is convention, as well as 50% power to allow direct comparison to previous work (e.g., Meyer et al. 2012). We indicate on all panels (both Figure 4 and Figure 4—figure supplement 1) the average number of gametes (n=1711) across donors in our study.

[Editors’ note: what follows is the authors’ response to the second round of review.]

Essential:– A comparison to hapcut2 seems to be crucial for benchmarking the method. Since your method has a phasing component, it is important to present a comparison that is adequate given the scale of the dataset.

We did not originally compare rhapsodi to HapCUT2 for two main reasons. First, HapCUT2 was developed as a diploid read-based phasing algorithm and is not compatible with the data structures and files used as input by rhapsodi. Second, we extensively compare rhapsodi to Hapi, a software package developed to impute sparsely sequenced gamete genotypes, and Hapi had been shown to outperform HapCUT2. However, given the comments from these Reviewers, as well as the adaptation of HapCUT developed and employed by our collaborator in Bell et al. (2020) for gamete sequencing data, we have revisited this topic for the revision. Note that while Bell et al. (2020) adapted the original version of HAPCUT, we instead adapted HAPCUT2 based on its improvements over the previous HapCUT algorithm (e.g., with regard to speed).

Applying HapCUT2 to the data input to rhapsodi requires a lengthy conversion process. We thus developed a file conversion pipeline (largely in line with that described by Bell et al. [2020]), with modifications to integrate with the other steps of rhapsodi. We then adapt the HapCUT2 algorithm for phasing, under the assumption that alleles originating from a single gamete and chromosome are linked. We offer this pipeline as an alternative phasing method within rhapsodi, called "linkedSNPHapCUT2". This pipeline is discussed in the Methods section "Reconstruction of phased donor haplotypes."

We compare the two phasing methods in rhapsodi – the default phasing approach termed "windowWardD2" and the HapCUT2 adaptation termed "linkedSNPHapCUT2." These comparisons are discussed in the Methods section "Comparison to existing methods'' (lines 1217-1237) and the Results section "Benchmarking against existing methods: Hapi and HapCUT2" (lines 297-454).

We applied "linkedSNPHapCUT2" to 5713 simulated datasets, on which it ran successfully in 77% of cases, but failed in 21% of cases based on the computing resources available (3 days runtime, 185Gb memory). Another 1% were unable to be processed due to extreme low coverages (88% below 0.001x) resulting in too few SNPs per chromosome (43 simulations with 1 SNP and 32 simulations with 2-5 SNPs). Of the simulations that failed with linkedSNPHapCut2, 43% were successfully phased with rhapsodi's default "windowWardD2" method. We compare completeness and accuracy between the two methods (Figure 3—figure supplement 3) as well as system/user and real time (Figure 3—figure supplement 4).

The major cost of the "linkedSNPHapCUT2" approach is the time and memory resources necessary to convert the input data files to the format necessary for use in HapCUT2. Because "windowWardD2" operates in windows along the chromosome, it runs as a multithreaded process. As such, its overall system time is larger than that of "linkedSNPHapCUT2" for datasets with high numbers of gametes, but wall-clock time may be much lower. Both options for phasing within rhapsodi offer high levels of completeness and accuracy across most data profiles (i.e., sequencing coverage and number of gametes), including those matching the Sperm-seq dataset. Overall, we find that in data limited scenarios, "linkedSNPHapCUT2" phases haplotypes with a higher completeness than the default "windowWardD2 method", but with comparable accuracy. For higher number of gametes, windowWardD2 offers comparable completeness and much higher accuracy, partially due to the HapCUT2 approach ignoring the positional information that was already encoded in the alignment. In doing so, this method does not take full advantage of the co-inheritance patterns of the SNP alleles, which is an advantage offered by the "windowWardD2" approach. Based on these results, we include both "windowWardD2" and "linkedSNPHapCUT2" in rhapsodi as alternative methods for phasing and recommend use of the latter in scenarios with small numbers of gametes and low coverage.

The above results are included in the text on lines 352-404, as well as in Figure 3—figure supplement 3 and Figure 3—figure supplement 4.

– The criticisms about pedigree-based studies need to be presented in a more nuanced way. Similarly, the broad statement about the absence of strong TD in humans seems poorly supported. The reviewers all suggested a more balanced presentation of these previous efforts and more of a discussion of how rhapsodi can be integrated in current research.

We have updated our presentation of previous efforts to investigate TD in humans (including pedigree-based studies), per the Reviewers’ specific comments below, in both the Introduction and the Discussion. These changes have better contextualized the nuances in statistical power, sample size, and allele frequency in each study design.

Specifically, we have clarified the power issues related to donor sample size and gamete size; we note that our study design differs from pedigree-based studies with regard to detecting TD affecting rare alleles. In our study, if an allele is rare in the population but heterozygous within just one donor, we are well-powered to detect TD. In pedigree-based studies, statistical power improves as the number of families heterozygous at a given site increases.

We further clarify our considerations of statistical power and allele frequency across populations: there is the issue of actually sampling the distorter allele, given its frequency in the population – a consideration that applies to both pedigree-based and gamete sequencing studies. Then, conditional on observing the allele, there is the issue of detecting TD given the sample size of informative transmissions. We have updated our statement regarding this topic (“single-gamete sequencing studies… provide equal power for detecting TD involving common and rare alleles”) to address this nuance.

We have addressed other differences between our study design and previous pedigree-based studies, including the possibility that pedigree studies capture TD or other effects occurring at other developmental timepoints, as well as the effects of selection (lines 579-583) and possible contributions of phenomena such as within-ejaculate sperm competition to signals of TD observed in other study designs that would not be detected in our current work (lines 724-743).

We also reworked our statement about the absence of strong TD in humans to more accurately reflect the evidence we have provided, which is specific to the sample under consideration. This begins with our statement of the TD result in the last paragraph of the Introduction and continues throughout the manuscript. In particular:

– We have added text to clarify that although TD occurring in repeat regions and segmental duplications would be removed through the filtering, we still expect to be able to detect the TD occurring across the chromosome, as evidenced by our simulations (lines 619-638).

– We address the possible cases of distorters masked or uncovered by admixture, and detection of multi-allele TD systems such as distorter and responder systems (lines 717-723).

Finally, we have expanded our discussion of how rhapsodi can be integrated in current research, facilitating novel uses of inferred genotype data and meiotic recombination events. These include:

– The ability to construct sex-specific genetic maps could provide essential genetic resources for non-model species to complement genome assemblies and annotations.

– The construction of personalized recombination maps for humans, which offers opportunities to investigate sources of variability in the recombination landscape, as well as its relation to phenotypes such as fertility status. For example, whole and partial chromosome gains and losses (i.e., aneuploidies) are the leading cause of human pregnancy loss, and one potential mechanism of aneuploidy formation is the abnormal number or placement of meiotic crossovers (Lamb et al. 2005).

– The opportunity for population genetic analyses using the chromosome-length haplotypes generated by rhapsodi, including inferring kinship and population structure, detecting signatures of positive selection, and assessing demographic history.

– The manuscript reads as two disjointed pieces, one on method development and the other on applications. We would ask that the manuscript be revised with this issue in mind, as it persists from the previous submission.

We have made significant updates to the manuscript, particularly following the suggestions from Reviewer 5, to better connect the method with its application and results. These changes have allowed us to reframe the paper, so that it more centrally focuses on our method and now features the scan for TD as a demonstration of an application of rhapsodi. We have added more clear transitions throughout all sections of the paper to improve the flow between use case, method, and application. We have also changed section headers to reflect this improved flow. As noted by multiple Reviewers, we believe these changes have improved the manuscript and will increase its readability and usefulness. Specifically:

– We have changed the title to more accurately summarize the work and appeal to potential readers and users by including the method in the title. It is now “A method for low-coverage single-gamete sequence analysis demonstrates adherence to Mendel's first law across a large sample of human sperm.”

– We have re-ordered the Abstract to highlight the need for new methods (such as rhapsodi) to analyze increasingly common datasets of sparsely sequenced gametes (such as SpermSeq). We then highlight TD in humans as an open question and one single gamete sequencing is well-suited to address. We then mention additional applications of rhapsodi.

– We have re-ordered the Introduction in a similar manner: starting with the availability of such datasets and the need for new methods to impute such sparsely sequenced data, we mention rhapsodi. We then move into the application to the SpermSeq data, outlining the state of the question of TD in humans currently and underscoring the opportunity offered by the SpermSeq data. Finally, we end with our findings regarding the absence of evidence of TD in this sample.

– We have applied these same changes to the Discussion: we begin with rhapsodi before introducing its application to the SpermSeq dataset and our findings regarding TD in that data.

Potentially useful but not essential:The reviewers list some suggestions, including changing the title, modifying the abstract, and reordering the introduction and discussion.Reviewer #2 (Recommendations for the authors):I appreciate the extensive further simulations, analyses, and revisions the authors have provided, particularly the additional simulations under strong TD and the clarified details about their samples, metrics, and computing resources. The suggestions for future uses of rhapsodi are also helpful in communicating the significance of this work. These additions substantially strengthen the manuscript and alleviate many of my initial concerns.

Thank you for considering our manuscript again at this step. We agree your previous comments strengthened the paper and look forward to further improvement from this second set of reviews.

I still feel that statements about generalizing a negative finding in the search for TD within this sample to a broader statement about the absence of strong TD in humans (e.g., line 107 "underscores the fidelity of human male meiosis for ensuring balanced transmission of alleles to the gamete pool"), are somewhat too strong, for the following reasons:1) All power simulations for rhapsodi-based TD detection are conditional on observing a distorter in heterozygous state within the sample, which would require the distorter not to be very rare (roughly >1.4% allele frequency required for 50% probability that at least one of 25 individuals is heterozygous). Observed distortion alleles in other species are often at very low frequency under an equilibrium model where the distortion is balanced by fitness costs in homozygotes (e.g., SD is found at 1 – 5% frequency per ref #9). Unbalanced distorters would fix or be lost rapidly, and would be unlikely to be detected except in the very brief window at which they are locally at intermediate frequency. The authors note these as caveats in the Discussion, but the need for a relatively common distorter is not described as an issue of power (see discussion of comparison to pedigree-based studies below).

We have addressed this and the next two reasons in detail throughout the manuscript. We have modified the above sentence to better reflect our finding of the absence of evidence of TD in this sample, as articulated by your comments here: “Our study thus suggests balanced transmission of alleles to the gamete pool in this sample.” While we acknowledge that TD detection is conditional on a donor being heterozygous for the distorter allele, our power analyses reflect the statistical power of our approach to detect TD if it occurs in a donor. We have added additional context addressing this limitation (the need for a relatively common distorter; lines 668-672) and added more detail regarding the comparison to pedigree-based studies, as outlined below.

2) I may have missed it, but I did not see mention of the possibility of population-specific TD aside from the discussion about fixed distorters uncovered through admixture. While it is true that there are very few alleles with extreme patterns of frequency differentiation across human populations (lines 620 – 621), this is not highly relevant for whether one should expect distorters to be population-specific. As mentioned above, distortion loci are likely to be rare and/or ephemeral, and either would make them very likely to be population-specific.

We agree that population-specific TD could be missed based on the limited diversity of this sample. We compared the genetic similarity of each of the 25 donor individuals in our dataset to globally diverse populations from the 1000 Genomes Project (see Figure 5 – —figure supplement 2). 16 individuals clustered with reference samples from the European superpopulation, 3 donors clustered with reference samples from the East Asian superpopulation, and 2 donors clustered with reference samples from the South Asian superpopulation. 3 donors clustered on an axis between the African and European superpopulations, and 1 donor showed similarities with both the African and East Asian populations. If distorters are rare or ephemeral, such distorters could be missed even in a diverse sample, which is a fundamental limitation of our study and other existing study designs for investigating TD in humans. We have added the additional caveat that population-specific distorters would likely not be included in our sample of 25 donors and discuss it in lines 709-713.

3) Some TD systems involve epistasis between alleles at distorter and responder loci, which would further require the responder allele to be observed heterozygous in the sample.

We agree that observing TD in such systems would require sampling an individual who is heterozygous at both loci. We would thus be less likely to capture this phenomenon, especially if the individual alleles are rare. More broadly, the possibility of missing rare alleles is a fundamental limitation of our study, but also of previous studies for detecting TD. Our study offers the advantage of requiring only one donor to be heterozygous for an allele for us to be well powered to detect TD, as opposed to other study designs that rely on aggregating data across multiple families and are thus more powered to detect TD involving more common alleles.

I did not notice this before, but there are statements in both the Introduction (line 63) and Discussion (line 505) about pedigree-based TD scans being underpowered due to the small size of human families. This is a bit misleading because such studies typically combine data across multiple families (as noted by the authors in lines 523 – 524), which reduces their power to detect TD present in any one individual but not to detect TD involving an allele at intermediate frequency that is heterozygous in many parents within the sample. Both pedigree-based and sperm sequencing studies have issues of power due to sample size of individuals that are currently discussed primarily in the context of pedigree-based studies; donor sample size concerns are separated from considerations of power when discussing the sperm sequencing study design.

We agree with the Reviewer that this is a nuanced issue. We have clarified the power issues related to donor sample size and gamete size, with changes to the paragraphs in the Introduction and the Discussion mentioned by the Reviewer. Specifically, we note that our study design differs from pedigree-based studies with regard to detecting TD affecting rare alleles: in our study, if an allele is rare in the population but heterozygous within just one donor, we are well powered to detect TD; in pedigree-based studies, statistical power improves as the number of families featuring the allele increases.

The statement that "single-gamete sequencing studies… provide equal power for detecting TD involving common and rare alleles" is undermined by the following clause "provided that they are heterozygous in the sampled donor individual." The probability that any of the sampled donor individuals are heterozygous for a distorter depends upon that distorter's frequency in the population, so the study's overall power to detect TD is not equal for common and rare alleles (see point 1 above). For example, in the context of Figure 4—figure supplement 4, 200 informative transmissions out of 1518 trios would imply p(heterozygous) = 0.066 (200/3036, MAF = 0.034). In the Sperm-seq sample of 25 individuals, 18% of such cases would be expected to have no heterozygotes in the sample. As this example demonstrates, the power for the present study is substantially higher than for previous pedigree-based studies, but not as much higher as implied by the power simulations.

We agree with the Reviewer that statistical power is affected by the distorter’s frequency in the population. We have chosen to consider these aspects of study design and power separately. There is the issue of actually sampling the distorter allele, given its frequency in the population a consideration that applies to both pedigree-based and gamete sequencing studies. Then, conditional on observing the allele, there is the issue of detecting TD given the sample size of informative transmissions. We have updated the sentence mentioned here to clarify this distinction and its impact on our study.

It might be worth considering for future work whether power may be gained for detecting TD using rhapsodi by combining data across individuals following haplotype inference.

Thank you for this suggestion. The idea of combining signals across donors (as is done with pedigree studies) is interesting and could potentially improve statistical power for common distorter alleles. However, it would require us to rework our approach for multiple testing correction (based on the effective number of statistical tests in light of LD). We will certainly consider this for future extensions of our work.

Reviewer #3 (Recommendations for the authors):Overall, the manuscript has improved and many of my concerns have been well addressed. In particular, emphasizing the power considerations of this specific approach in the main text was an important addition. I appreciate the authors' correcting inaccuracies in my understanding of their previous work.I respectfully disagree with the authors' dismissal of several considerations.First, the paper remains a pretty sharp subdivision of methods vs biology. The methods are really about phasing and recombination detection in sperm-seq data, the biology is about TD. I believe it will be hard for a non-specialist to read this manuscript, though the authors are correct that extremely specialized users --- i.e., those with their own sperm-seq datasets --- may benefit from improved software usability.

Thank you for this feedback on the organization of the paper. We agree with the need to generalize the paper and make it more applicable and useful to non-specialist readers. We have made extensive changes to address this, including implementing many suggestions shared by Reviewer 5. Specifically, we updated the title to encapsulate both the method and its application; re-ordered the Abstract, Introduction, and Discussion (to first articulate the need for a new method/software to analyze low-coverage sequencing data from a large sample of haploid gametes; highlight TD in humans as an example of a question that can be addressed with the SpermSeq data more effectively than previous datasets; and summarize our findings); and added explicit mentions of other applications of rhapsodi to the Discussion. These changes are detailed in our response to essential revisions above.

Second, a comparison to an approach that is suited to the scale of data used is necessary. If hapcut2 is the only option, it should be applied despite being an "off-label" use. Also, yes, it is somewhat outside of typical hapcut2, but linking the reads bioinformatically is pretty straightforward and reasonable. It bears some similarity to read-backed phasing Certainly if one of this studies' co-authors believed this to be a valid use in previous published work it is reasonable to include as a point of comparison here.

As outlined in our response to essential revisions above, we have developed a pipeline to convert the input files from rhapsodi into a format usable in HapCUT2 and then call the phasing algorithm from HapCUT2 from within the rhapsodi package. We now include this as an alternative phasing approach in rhapsodi called `linkedSNPHapCUT2.` We compare this phasing approach with the default phasing option in rhapsodi (`windowWardD2`) for accuracy and completeness (Figure 3—figure supplement 3) and for system/user and real time (Figure 3—figure supplement 4) in simulated datasets (5,713 and 216, respectively) generated across a range of parameter spaces (coverage ranging from 0.001x to 2.303x and number of gametes ranging from 3 to 5,000).

On a related note, in "Benchmarking against existing methods", while the description of previous analysis is accurate, the relevant underlying datasets vary so much from previous works that is it not clear that the same results would hold here. Minimally, some acknowledgement should be added that the results described from previous work are based on a dramatically different dataset than was considered here.

We have added text (line 299-305) at the beginning of this section acknowledging that the results described from previous work are based on a dramatically different dataset than was considered here and clearly outlining our selection of Hapi for downstream comparisons. We also address this by comparing the default phasing method in rhapsodi, `windowWardD2`, with the modified phasing from HapCUT2, `linkedSNPHapCUT2` on the same simulated datasets.

Reviewer #4 (Recommendations for the authors):The authors describe two main things in this paper:First is the software package called "rhapsodi". Rhapsodi is an R package which is designed to use sparse whole-genome sequencing data from gametes to phase a sperm donor's haplotypes, impute gamete genotypes and identify meiotic crossover breakpoints.Second, rhapsodi was used with both simulated data and a published dataset of 41,189 individual sperm cells' sequence. The main biological focus of this second part of the paper involved testing for transmission distortion, of any alleles or broader linkage blocks, for which none is detected. Previous publications, referenced in the manuscript, have suggested that transmission distortion can occur in human populations. The possible causes of the discrepancies in results between the study and previous studies is discussed.StrengthsIn the set of variables assessed, rhapsodi appears to outperform Hapi, which the authors present as the benchmark to meet or exceed for a package focused on efficient, accurate, complete and reliable phasing of haplotypes.Another strength is that the paper also assesses transmission distortion with particularly large datasets for which no signal is detected. This contrasts from previous studies cited in the manuscript. However, the claims relevant to the analysis of this public dataset appears to be convincing. Further, the authors go on to offer good reasoning, particularly in the Discussion, about how they arrived at these conclusions, what any potential limitations of the approach may be, and how the dataset used differs fundamentally from pedigree-based data i.e. this study investigates transmission rates in gametes and does not consider different stages of fertilisation and subsequent aspects embryonic development.

Thank you for acknowledging the contributions of our work, and for your close reading of our paper.

WeaknessesThe approaches in the paper generally are quite strong. However, the study lacks a positive control for transmission distortion, which is not simulated data. This positive control does not exist in published human datasets to the best of my knowledge. However, outside of the scope of this study, one could consider reanalysing published non-human single-gamete sequencing datasets of which there are a number of studies.

We agree that a positive control of a non-human example of TD in single-gamete data would be a useful demonstration, though we agree that it is outside the scope of the current manuscript.

The analyses of meiotic crossovers with non-simulated data are quite light – limited to Figure 5 sup 4 and 5 – compared to the two other features of rhapsodi (phasing and imputation), which feature more extensively. Given that a significant portion of the manuscript presents a re-analysis of the data from Bell et al. 2019 Nature, it be helpful to visually demonstrate the variation in crossover numbers, positioning, interference between the 25 donors, and where possible compare these results to published analyses e.g. Bell et al. and other studies. For example, Figure 5—figure supplement 5 suggests that rhapsodi is much more conservative with calling of crossovers/haplotype transitions than Bjarni et al. and I think that this is worthy of discussion. Particularly because false positive rates of double crossovers can have a large effect on particular study types, such as those concerning crossover interference.

Thank you for acknowledging the contributions of our paper. We feel our analysis and discussion of meiotic crossovers is sufficiently detailed, supported by the two figures with non-simulated data mentioned by the Reviewer (Figure 5 – —figure supplements 4 and 5). Our method, rhapsodi, is a thoroughly documented and accessible software package for phasing, imputation, and identifying recombination hotspots. Our current manuscript focuses on phasing and imputation, which are the key steps that enable downstream applications of rhapsodi, including the scan for transmission distortion (which is novel and not investigated in Bell et al. 2020). Further investigating the crossovers of these donors (including in comparison to work by Bell et al.) is beyond the scope of the current paper; indeed, topics including the analysis of inter-individual variability in the crossover landscape were the main focus of Bell et al. (2020).

As described in our responses to previous reviewer comments, one such technical reason for the observed modest differences between our findings and those from Bjarni et al. appears to be related to the sample sequencing depth of coverage. Rather than pooling the number of inferred recombination events for each bin across all donors, we repeated the correlation analysis in a donor-specific manner. We then fit a linear regression model with the donor-specific sequencing depths of coverage as the predictor and the donor-specific correlations as the response variable. We found that the donor-specific correlation with the deCODE map was positively associated with depth of coverage (lines 463-470).

The outputs of this work will be very useful for future researchers; both the rhapsodi software, and the finding that there is no strong transmission distortion signal in these 25 male sperm samples. Rhapsodi is one of a very limited number of user-friendly generalised pieces of software – as opposed to a collection of scripts – for the analysis of sparsely sequenced gametes for haplotype phasing, imputation and meiotic crossover breakpoint analysis. The package should attract significant attention from the potential userbase.

Thank you for your enthusiasm about our software and findings.

The biological findings of this study – failure to identify transmission distortion in these samples – should be of general interest to geneticists and adjacent fields. Despite being a negative finding, it challenges existing literature with a robust analysis and will be sure to stimulate further research directions.

Thank you for your comment on our findings.

I found the paper very interesting. I also found the paper to be very well written, clear, accessible to a broad readership, and look forward to using rhapsodi.Reviewer #5 (Recommendations for the authors):The authors introduce a new method rhapsodi, which accurately infers haplotypes from low-coverage sequence data of large sample sizes of haploid gametes. They demonstrate that rhapsodi performs better than the existing approach Hapi, particularly for larger samples of gametes. They apply rhapsodi to test for evidence of transmission distortion (TD) in a published human Sperm-seq dataset (single cell sequencing of >30k sperm from 25 donors), and report no significant evidence of TD.rhapsodi is a powerful approach, which will be useful in a variety of contexts. The TD analysis is more rigorous than previous pedigree-based studies, providing convincing evidence for a lack of strong distorters. However, power to detect weak TD remains limited after accounting for multiple testing. The method is a larger contribution, and likely to be of interest to a broader audience. The manuscript would benefit from re-framing to focus on introducing rhapsodi – with human TD results demonstrating its application.

Thank you for your comments, and for your close reading of our work. We agree that reframing the manuscript has helped emphasize the contributions of the Method and have addressed your feedback to that end, as detailed in the response to essential revisions above. We walk through those updates in response to your individual comments below.

Recommendations for authors:The authors have thoroughly and carefully addressed the technical concerns of previous reviewers, and better placed the results in context with previous literature.However, as noted by previous reviewers, the major strength of the manuscript is the method rather than the human TD analysis. Re-framing the manuscript with focus first on the method would better reflect the relative contribution of the aims, and broaden readership. This could be accomplished without extensive re-writing. For example, I suggest:– Change title: The title refers only to the TD results – including mention of the method in the title will be more accurate and make it less likely to be overlooked by readers interested in applying the method to other questions. e.g. something like:New method for analyzing haploid gamete sequences applied to a large sample of human sperm shows adherence to Mendel's first law.

Thank you for the title suggestion. We have reworded it slightly, keeping the mention of the method and the suggestion of its other applications: “A method for low-coverage single-gamete sequence analysis demonstrates adherence to Mendel's first law across a large sample of human sperm.” We agree that this more accurately summarizes our findings and emphasizes the method, as suggested by this Reviewer and by Reviewer 3.

– Re-ordering abstract: to first highlight the need for new method/software to analyse SpermSeq data – ie low-coverage sequencing data from large sample size of gametes. Next highlight TD in humans as an example of a question that can be addressed with SpermSeq data better than previous. [summary of findings] Add at the end more explicit mentions of other applications of rhapsodi (outlined in L653-672 of Discussion)

Thank you for the specific suggestions, which we agree have strengthened our paper. We have re-ordered the Abstract to highlight the opportunity for new methods to analyze increasingly common datasets of sparsely sequenced gametes (such as SpermSeq). We then highlight TD in humans as an open question and one single gamete sequencing is well-suited to address. We then mention additional applications of rhapsodi.

– Re-order Introduction/beginning of Discussion in similar manner.

We have re-ordered the Introduction in a similar manner: starting with the availability of such datasets and the need for new methods to impute such sparsely sequenced data, we mention rhapsodi. We then move into the application to the SpermSeq data, outlining the current knowledge regarding TD in humans and underscoring the opportunity offered by the single-cell gamete data such as SpermSeq. Finally, we end with our conclusion regarding the absence of evidence of TD in this sample.

We have applied these same changes to the Discussion: we begin with rhapsodi before introducing its application to the SpermSeq dataset and our findings regarding TD in that data.

Other comments:L402 "even slight deviations from Mendelian expectations, as supported by our simulations of TD across a range of gamete sample sizes and transmission rates"– I would consider ~50-55% slight deviations – power is low in this range– Single noted peak on Chr 2 is 56% for 1571 gametes – but doesn't reach genome-wide significance– This concern has been addressed somewhat by removing "strict" etc in response to previous reviewers, but could

We acknowledge that the statement regarding slight deviations is open-ended, especially in light of the other information in that paragraph and results we present in our study. As such, we have added additional notes in the text regarding the power in the range of 50-55% deviations.

L491 comparison to Zollner – excess of allele sharing among sibs in pedigree – 50.43% -this is surviving offspring not gametes so reflects additional sources of TD and selection on embryos/offspring

Thank you for illustrating this important point that pedigree-based studies and gamete-based studies focus on data at completely different developmental stages and are thus subject to numerous different factors. We have added a sentence after this line to note that Zollner et al. and other pedigree studies are based on surviving offspring (rather than gametes) and thus may reflect additional sources of TD as well as selection and other factors.

L621-631 discusses other sources of TD but does not mention this may explain the discrepancy with pedigree based approachesCould add within-ejaculate sperm competition as factor – reviewed in:https://royalsocietypublishing.org/doi/full/10.1098/rstb.2020.0066#d1e627

We agree that the other factors (e.g., selection, physiology) affect samples between the study timepoints of gamete-based studies (i.e., pre-fertilization) and pedigree-based studies (i.e., two-generation family trios). As such, these factors can explain the discrepancy between our finding of balanced transmissions in the gamete pool and past findings of TD in pedigrees. We now mention this in the Results section “No global signal of biased transmission in human sperm” (lines 579-583). We have added text in the Discussion after listing these factors and specifically state that these can explain the discrepancy between the results and conclusions from past work and our current study: “These other forms of TD (in addition to other factors occurring throughout development between the timepoints of gamete-based studies such as ours and pedigree-based studies) can explain the differences in the conclusions of these studies” (lines 723-732). We have also added a sentence in this section mentioning that within-ejaculate sperm competition may contribute to signals of TD observed in other study designs (such as pedigrees) but would not be detected in this study of a pool of gametes, and have included the citation recommended here (lines 733-736).

L581 this paragraph makes more sense combined with points in L543-550

We agree that the paragraph referred to (L581, beginning with “One caveat of this and most previous scans for TD in humans”) makes more sense combined with the other points mentioned; we have thus combined that paragraph with that referred to by L543-550.

Filtering of repeat regions and segmental duplications could remove sites with TD (as noted by the previous reviewer) – in L545 point out the benefits of stringent filtering but doesn't point out that some of these regions could be effected by TD

We have added text to clarify that although TD occurring in these regions would be removed through the filtering, we still expect to be able to detect the TD occurring across the chromosome, as evidenced by our simulations (lines 615-625).

Methods – in beginning, briefly reiterate the differences compared to methods in the sperm-seq paper (Bell et al. 2020) – it's unclear that the method here is similar but formalised into the package

To clarify the differences between rhapsodi and the methods in Bell et al. 2020, we have added an additional paragraph to the Methods subsection “Comparison to existing methods.” In this we explain the differences in the phasing and crossover detection steps, and emphasize that rhapsodi is a formalized package, as noted by this Reviewer (lines 1238-1250).

Figure 4 – Figure supp 1 and Figure 2-supp9 – can't distinguish different colors – make points bigger and/or change shape also according to scale

Thank you for pointing out the issues with these figures. We have increased the size of the points in both figures to increase visibility.